# Differential effects of intra-modal and cross-modal reward value on perception: ERP evidence

**Roman Vakhrushev[1], Felicia Pei-Hsin Cheng[1], Anne Schacht[2], Arezoo Pooresmaeili** [1] *

**1** Perception and Cognition Lab, European Neuroscience Institute Goettingen- A Joint Initiative of the University Medical Center Goettingen and the Max-Planck-Society, Goettingen, Germany, **2** Affective Neuroscience and Psychophysiology Laboratory, Georg-Elias-Müller-Institute of Psychology, Georg-August University, Goettingen, Germany

* a.pooresmaeili@eni-g.de

**Data Availability Statement:** The data and analysis scripts that support the findings of this study are uploaded to the Open Science Framework data repository (https://osf.io/47wxr/files/osfstorage#).

## Abstract

In natural environments objects comprise multiple features from the same or different sensory modalities but it is not known how perception of an object is affected by the value associations of its constituent parts. The present study compares intra- and cross-modal value-driven effects on behavioral and electrophysiological correlates of perception. Human participants first learned the reward associations of visual and auditory cues. Subsequently, they performed a visual discrimination task in the presence of previously rewarded, task-irrelevant visual or auditory cues (intra- and cross-modal cues, respectively). During the conditioning phase, when reward associations were learned and reward cues were the target of the task, high value stimuli of both modalities enhanced the electrophysiological correlates of sensory processing in posterior electrodes. During the post-conditioning phase, when reward delivery was halted and previously rewarded stimuli were task-irrelevant, cross-modal value significantly enhanced the behavioral measures of visual sensitivity, whereas intra-modal value produced only an insignificant decrement. Analysis of the simultaneously recorded event-related potentials (ERPs) of posterior electrodes revealed similar findings. We found an early (90–120 ms) suppression of ERPs evoked by high-value, intra-modal stimuli. Cross-modal stimuli led to a later value-driven modulation, with an enhancement of response positivity for high- compared to low-value stimuli starting at the N1 window (180–250 ms) and extending to the P3 (300–600 ms) responses. These results indicate that sensory processing of a compound stimulus comprising a visual target and task-irrelevant visual or auditory cues is modulated by the reward value of both sensory modalities, but such modulations rely on distinct underlying mechanisms.

## Introduction

Reward seeking is a fundamental mechanism for survival and one of the main predictors of behavior [1, 2]. We tend to prioritize what we eat, where we go and what we do based on the expected value of objects or actions learned through experience. A large body of literature has

Further information regarding the data and the analysis scripts can be obtained from the corresponding author (AP).

**Funding:** This work was supported by an ERC Starting Grant (no: 716846) to AP. The funders had no role in study design, data collection and analysis, decision to publish, or preparation of the manuscript.

**Competing interests:** The authors declare no competing interests.

identified a network encompassing the ventral striatum and orbitofrontal cortex to play a key role in learning the associated value of neutral stimuli through experience [3, 4]. As natural environments are rich and dynamic, reward information could be conveyed through multiple sensory modalities (e.g., hearing the sound of an approaching ice-cream truck or seeing its characteristic red color both inform us of the possibility of enjoying an ice cream) and reward associations or the goals of the task at hand may change over time (e.g., a red truck can carry other items rather than ice-cream). These features entail a tight interaction between the reward network and the early sensory areas so that stimuli leading to better outcomes such as higher rewards or realization of the goals of the task are prioritized for perceptual processing [5, 6]. In fact, previous research has identified value-driven modulations of neuronal responses in almost all primary sensory areas [7–12]. In line with the effect of reward on the earliest stages of sensory processing, studies on humans with the use of electroencephalography (EEG) reported reward-related modulations of the visual event-related potentials (ERPs) that are likely to originate from the primary and extrastriate visual cortices [but see also 13], including modulations in P1 (80–120 ms) [14–16], N1 (140–220 ms) [17] and C1 (~70 ms) [18, 19] ERP components. Despite the robust evidence for an influence of reward on early sensory mechanisms, especially in the visual cortex, it has remained unclear how reward information and sensory processing are coordinated across multiple sensory modalities. This is an important question as in natural environments stimuli are typically multisensory and reward information should be hence coordinated across multiple sensory modalities.

Two recent studies tried to bridge this gap and examined cross-modal reward mechanisms, where an auditory stimulus associated with positive reward value affects visual perception [20, 21]. During a conditioning phase, participants associated different pure tones with different reward values. In a subsequent post-conditioning phase, participants either reported the location [20] or the orientation [21] of a near-threshold Gabor stimulus in the presence of task-irrelevant auditory tones. Importantly, during the post-conditioning phase, auditory tones did not predict the delivery of reward anymore. The first study [20] showed that participants' accuracy in determining the location of the visual target was higher in the presence of tones previously associated with a positive reward compared to no reward (neutral). The authors concluded that the tones associated with rewards enhanced the bottom-up salience of the visual stimuli. Similar results were found in the second study [21], as it was shown that participants' orientation discrimination improved in the presence of auditory tones previously associated with high compared to low reward value. With the aid of the simultaneously acquired functional MRI data, it was further shown that the enhanced orientation discrimination is also reflected in the activation patterns of the early visual areas elicited by a specific stimulus tilt orientation [21]. Moreover, in addition to the classical reward coding areas (such as Striatum and Orbitofrontal cortex), multisensory regions of the temporal cortex were also modulated by sound values suggesting that they may serve as an intermediary stage to better coordinate the interaction between the primary sensory cortices when high-value stimuli were presented [21]. Subsequent studies [22–25] used a variety of tasks encompassing visual search or detection or discrimination tasks and confirmed that cross-modal (auditory) reward cues could affect visual perception, albeit the cross-modal reward cues in some cases interfered with the visual task [22, 25] and in other cases improved it [23, 24]. Together, these results provide evidence that reward effects can occur cross-modally. However, it remains unclear whether the modulatory effects of reward on perception depend on whether reward cues are from the same or different sensory modality as the task-relevant target stimulus.

A predominant view posits that reward effects on sensory perception occur through the engagement of attentional mechanisms [26–29]. In this view, rewarding cues receive higher attentional prioritization either through an involuntary, value-driven attentional capture [30],

through voluntary, goal-directed attentional selection [31–34], or through cognitive control mechanisms [35, 36]. In line with this, it has been shown that a reward cue that is aligned with the goal-directed attention in space and in time improves visual performance [31, 37], whereas when the rewarded stimulus is presented away from the target position, it interferes with the task [38, 39]. Considering these findings, enhancement of visual perception by co-occurring sounds [20, 21] is unexpected, as auditory tones in these studies were irrelevant to the visual task and could potentially act as a high-reward distractor capturing attention away from the visual target. One possibility is that cross-modal reward cues enhance visual perception by strengthening the audiovisual integration of the auditory and visual components of an audio-visual stimulus. Although audiovisual integration largely occurs automatically [for a review see 40], top-down factors such as attention [41–43] and recently reward value [44, 45] have been shown to affect its strength. Through a more efficient integration with the visual cues, auditory reward signals could hence capture attention not only to themselves but also to the whole audiovisual object including the visual target [46], thereby improving performance. In fact, the boost of integration may be a key characteristic of reward modulation where the association of one sensory property of an object with higher reward spreads to all sensory properties of the same object hence promoting their grouping, as has been shown before in the visual modality [11]. Therefore, although previous research has provided evidence for a spread of reward-driven effects across different parts of visual [11] and audiovisual objects [20, 21], these effects have never been compared against each other.

The central aim of this study was to provide a better understanding of reward effects across sensory modalities by comparing intra- and cross-modal reward effects, while using the high temporal resolution of the EEG data to delineate the different stages of stimulus processing across time. To this end, we employed a task design similar to a previous study [21], where participants first learned the reward value of visual or auditory cues during a conditioning phase and subsequently performed a visual orientation task in the presence of previously rewarded cues. During the latter post-conditioning phase, auditory and visual reward cues were presented simultaneously with the target stimulus and at the same side of the visual field. Simultaneous presentation of reward cues and visual target allowed us to test whether the reward associations of task-irrelevant parts of a compound object (i.e. previously rewarded auditory and visual cues) can affect the processing of the task-relevant visual target. In this case, the spread of value-driven modulations from the visual or auditory cues to the target will indicate object-based effects in the visual modality [47] or cross-modally [46]. However, since this design compared visual and audiovisual stimuli that are known to elicit different magnitudes of neural responses [48], intra-modal and cross-modal high reward conditions were contrasted with their low reward counterpart from the same sensory configuration, thus allowing us to isolate the reward-driven effects independent of the sensory configuration of stimuli. Crucially, reward cues during the post-conditioning phase were *irrelevant to the task* at hand and *did not predict the delivery of reward* (i.e. no-reward phase). Measuring reward effects at a no-reward phase has proven to be an effective method to separate different modulations of perception, i.e., those related to the long-term associative value of reward cues from those of goal-directed boosts driven by reward-predicting cues or the delivery of the reward itself [49–54].

We hypothesized that cues previously associated with high value should positively influence behavioral performance (i.e., enhancing target's discriminability and reducing the reaction times) and early posterior ERP components (i.e., increasing the amplitudes of P1 and N1 components, and decreasing their latencies). This hypothesis is based on a mechanism in which value-driven prioritization of task-irrelevant reward-associated cues spreads to other sensory components of the same object and thereby enhances the representation of the target irrespective of the sensory modality of the reward cue. Next, we hypothesized that reward-related

modulations of P1 and N1 components in posterior electrodes occur earlier and are stronger when the reward cue is delivered intra-modally (in visual modality) compared to when the reward cue is delivered cross-modally (in auditory modality), as intra-modal effects rely on direct neuronal connections between reward and target sensory representations [55, 56], whereas cross-modal effects rely on the long-range communication between different brain areas and/or involve intermediate stages [21]. Finally, we hypothesized that in later stages of information processing (i.e. > 250 ms), intra- and cross-modal reward cues elicit similar modulations of ERP amplitudes, both leading to an enhancement of P3 ERP component.

## Material and methods

### Participants

Thirty-eight participants took part in our experiment. Two participants were excluded since their performance during the associative reward learning task indicated that they either did not learn the reward associations (N = 1) or had difficulties with discriminating the location of the stimuli (N = 1). The final sample consisted of 36 healthy participants (23 women, mean age ± SD: 25.8 ± 5 years; 28 right-handed) with normal or corrected-to-normal vision who had no history of neurophysiological or psychiatric disorders according to a self-report. Participants gave written informed consent, after the experimental procedures were clearly explained to them. The study was conducted in full accordance with the Declaration of Helsinki and was approved by the local Ethics Committee of the Medical University Göttingen (proposal: 15/7/15).

The sample size, all procedures, and the analysis plan of the study were preregistered (https://osf.io/47wxr/). The required sample size was calculated based on a pilot study (N = 8) indicating that at least 33 subjects were needed to detect a significant difference between high and low reward value ($\alpha = 0.05$, $1 - \beta = 0.8$; GPower [57]). To maintain the counterbalancing of our experimental conditions (4 possible combinations of auditory and visual cues with high or low reward), we aimed for an a priori sample size of N = 36 before the data collection started.

### Experimental procedures

Data collection was done in a darkened, sound-attenuated, and electromagnetically shielded chamber. Participants sat 91 cm away from a 22.5-inch calibrated monitor (ViewPixx/EEG inc., resolution = 1440 × 980 pixels, refresh rate = 120 Hz) with their heads resting on a chinrest. Stimulus presentation was controlled in Psychophysics toolbox-3 [58] in MATLAB (version R2015b) environment. Eye movements were recorded with an Eyelink 1000 eye tracker system (SR Research, Ontario, Canada) in a desktop mount configuration, recording the right eye at a sampling rate of 1000 Hz. Electrophysiological data were recorded from 64 electrodes (BrainVision Recorder 1.23.0001 Brain Products GmbH, Gilching, Germany; actiCap, Brain Products GmbH, Gilching, Germany), online referenced to TP9, digitized at 1000 Hz, and amplified with a gain of 10,000. Electrode impedances were kept below 10kΩ.

Each participant performed two tasks: an associative reward learning task (conditioning phase) (**Fig 1A**) and a visual orientation discrimination task in the presence of task-irrelevant auditory or visual cues (pre- and post-conditioning phases, **Fig 1B**). An experimental session started with a calibration procedure, where for each participant the luminance of two consecutively presented colors was adjusted until the perceived flicker between them was minimized and they became perceptually isoluminant (total duration of calibration < 5min). The calibration was followed by a short training session for the orientation discrimination task (Number of trials = 36). After training, each participant's orientation discrimination threshold was determined using the QUEST method (an adaptive psychometric procedure that selects the

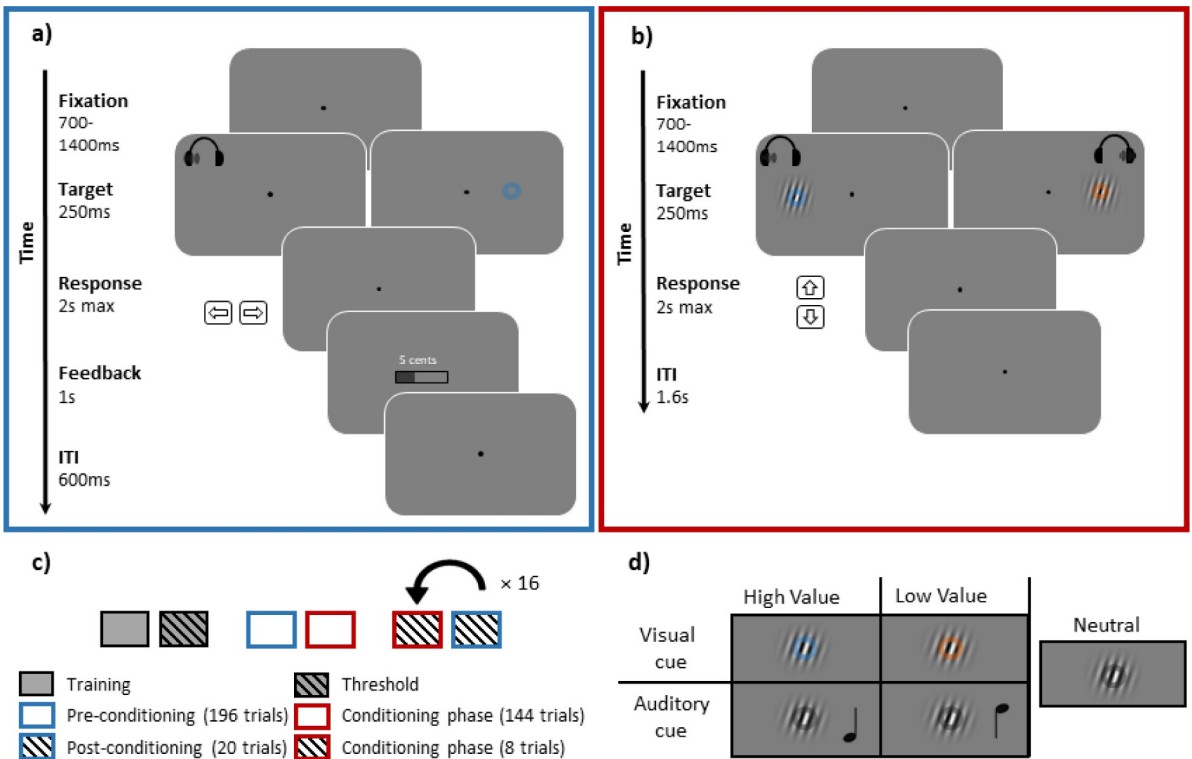

**Fig 1. Stimuli and experimental procedures. a) Reward learning phase (conditioning).** Participants learned the reward association of the visual or auditory cues by performing a localization task and observing their monetary rewards contingent on the color (visual) or pitch (auditory) of the cues. In this task, after an initial fixation period (700–1400 ms), a visual or an auditory cue was presented to the left or right side of the fixation point and participants had to localize them by pressing either left or right arrow buttons of a keyboard (maximum response time 2 s). Here, stimuli in two example trials are shown, one with an auditory cue presented to the left and the other with a visual cue presented to the right. The correct performance led to either a high or a low monetary reward dependent on the identity of the cue (color of visual cues and pitch of auditory cues). **b) Orientation discrimination task employed during the pre- and post-conditioning phases.** To probe the effects of reward value on visual sensitivity, an orientation discrimination task was employed. A trial of this task started with a fixation period (700–1400 ms), followed by the presentation of a peripheral Gabor stimulus (9° eccentricity). Participants were instructed to discriminate the orientation of the Gabor stimulus (clockwise or counterclockwise tilt) by pressing down or up arrow buttons on a keyboard (maximum response duration = 2 s). Concurrent with the Gabor, a visual or auditory cue was also presented (intra- or cross-modal cues, respectively). Intra- and cross-modal cues were irrelevant to the orientation discrimination task and did not predict reward delivery. **c)** The schematic illustration of the different stages of an experimental session. Note that after a long conditioning block where participants learned the stimulus-reward associations, smaller blocks of conditioning were interleaved with the blocks of post-conditioning (16 repetitions) to prevent the extinction of reward effects. **d)** Design matrix of stimulus conditions used during the pre- and post-conditioning phases to assess the effect of reward (high or low) and sensory modality (intra- or cross-modal). A neutral condition was also included during the pre- and post-conditioning phases, which was never associated with any reward value and served to assess the responses evoked by the visual target. Note that the reward assignments of visual and auditory cues were counterbalanced across participants.

stimulus intensity- here the tilt orientation- on each trial at the current most probable Bayesian estimate of threshold [59]). This procedure was adjusted to find a tilt degree where a participant reached an accuracy level of 70% as their perceptual threshold. Thereafter the session proceeded to the pre-conditioning, conditioning and post-conditioning phases (**Fig 1C**).

During pre- and post-conditioning phases, participants were instructed to report the tilt orientation of a Gabor stimulus relative to the horizontal meridian by pressing a keyboard button (down and up arrow keys for clockwise and counterclockwise tilts, respectively). Gabor stimuli were Gaussian-windowed sinusoidal gratings with an SD of 0.33°, a spatial frequency of 3 cycles per degree, 2° diameter, and 50% contrast, displayed on a gray background of 34.4 cd/m² luminance, presented at 9° eccentricity. A transparent ring (0.44° in diameter, 0.17 pixels thick, alpha 50%) was overlaid on the Gabor in all trials. The color of the ring depended on the

experimental condition (**Fig 1D**): blue or orange colors were paired with different reward magnitudes during the conditioning phase and are referred to as intra-modal reward cues, whereas the grey color was never paired with rewards and was used in cross-modal and neutral conditions (color in HSV = [20, 0.4, V] for orange, [250, 0.4, V] for blue, and [0, 0, V] for grey, where V was calibrated for each participant to produce isoluminance). Auditory cues were pure monaural tones, lateralized to the left or right earphone (sound level pressure = 70 dB, delivered through earphones: ER-3C, Etymotic Research Inc.) and were presented simultaneously with the Gabor target. In cross-modal conditions, different tone frequencies (350 Hz or 1050 Hz) were associated with either high or low reward during the conditioning. Therefore, in our design only one feature of a stimulus, either the color of the ring in intra-modal stimuli or the pitch of the tones in cross-modal stimuli, had a distinct pairing with a high or low reward, whereas other features (a Gabor and an overlaid circle) were shared among all stimuli. Participants were instructed to focus on judging the orientation of the Gabor and ignore the task-irrelevant visual and auditory cues, and were additionally told that correct responses in the orientation discrimination task granted them a fixed reward shown at the end of the experiment. These instructions aimed to emphasize that visual and auditory cues were not relevant to the task and did not lead to differential immediate rewards after conditioning. In total, participants performed 160 trials in the pre-conditioning phase and 320 trials in the post-conditioning phase, with 32 and 64 repetitions of each of the value (high, low) × modality (intra-modal, cross-modal) conditions, as well as the neutral condition.

During the conditioning phase (**Fig 1A**), participants reported the location of a peripheral visual (9˚ eccentricity) or auditory stimulus (sound played in the left or right earphone) by pressing left or right arrow keyboard buttons. Visual and auditory cues had the same characteristics as in pre- and post-conditioning. However, note that the neutral condition included during the orientation discrimination task, was not presented to the participants during the conditioning and was therefore not associated with any reward value. Correct responses were followed by a monetary reward, the magnitude of which depended on the respective color or pitch of the visual or auditory cues (in case of an error the monetary reward was set to zero). Reward information was presented at the screen center both in written format (with digits showing the number of Euro cents obtained) and graphically (a color bar illustrating the ratio of the reward magnitude in current trial to the maximum possible level in the experiment, respectively), and stayed in view for 1 s. Rewards were drawn from two Poisson distributions with the mean of 25 cents and SD of 4.8 cents for high-reward trials and a mean of 2 cents and SD 1.4 cents for low-reward trials (minimum and maximum reward was fixed at 35 and 0.4 cents, respectively). Participants completed 280 trials of the cue localization task to learn the reward associations. From these, 144 trials were presented during an initial longer conditioning block (comprising 36 repetitions of value × modality conditions). Subsequently, 136 trials of the conditioning phase were divided into 16 short blocks (8 trials per block) and then interleaved with short blocks of the orientation discrimination task during post-conditioning (20 trials per block) to prevent the extinction of the reward associations (**Fig 1C**). Participants were instructed to remember and report the color and the sound pitch that delivered higher rewards at the end of the conditioning phase and at the end of the experiment (by indicating which of the two sequentially presented colored circles or auditory tones had higher rewards).

Throughout the experiment, trials were repeated in case participants made an eye movement (> 2˚ displacement of gaze from the fixation dot during fixation or target period), used an undesignated button to respond, or did not respond at all to maintain equal number of repetitions across conditions. The location (left or right), modality (auditory or visual), and identity (color or sound pitch) of the cues were pseudorandomized across trials with the constraint that the same condition could not be repeated more than twice in a row.

The association of each cue identity with the reward value was counterbalanced across participants. This means that each reward condition (high or low), comprised an equal number of instances where either the orange or the blue color or the 350 Hz or the 1050 Hz tone was associated with that reward magnitude. Therefore, when describing the effects during the orientation discrimination task, we will refer to all stimulus conditions with respect to the reward assignment that they acquired after the conditioning, although during the pre-conditioning these associations were not learned yet. Accordingly, these conditions in both pre- and post-conditioning will be referred to as: **I**ntra-modal **H**igh Reward: **IH**, **I**ntra-modal **L**ow Reward: **IL**, **C**ross-modal **H**igh Reward: **CH** and **C**ross-modal **L**ow Reward: **CL**, plus a neutral condition (referred to as Neut), which was never associated with any reward value (as shown in **Fig 1**). Also note that the neutral condition comprised a Gabor patch and a semi-transparent ring overlaid on it, a feature that was shared among all stimulus configurations. Since this condition was never associated with any reward, it served as a means to characterize the behavioral and neural responses to the visual target, independently from the responses to reward-associated features of the stimuli (i.e. colors and sound pitches).

## Data analysis

The main focus of our study was on the value-driven effects on behavior and ERPs. However, following our pre-registered plan, we also examined the effect of rewards on the pupil size and measured the correlation between reward effects and participants' scores on a standard reward sensitivity test and these results are reported in S1 Text.

**Analysis of the behavioral data.** For behavioral assessment of visual sensitivity during the orientation discrimination task, we used participants' d-prime scores (d'). D-prime was measured based on the probability of hits and false-alarms, as d' = $Z(P_{Hit})$—$Z(P_{FA})$, where one of the tilt directions was arbitrarily treated as "target-present" as in formal Signal Detection Theory analysis of discrimination tasks [60, 61]. Extreme values of $P_{Hit}$ or $P_{FA}$ were slightly up- or down-adjusted (i.e., a probability equal to 0 or 1 was adjusted by adding or subtracting 0.001, respectively). Reaction times in all phases of the experiment were calculated as the elapsed time between the onset of the target stimulus and the participant's response. The resulting response times were averaged for each experimental condition (including correct and incorrect responses).

Outliers were removed from the behavioral and ERP data of each participant (0.38% ± 0.6 SD of all trials across all subjects). A trial was considered to be an outlier if the gaze fixation during the target presentation period was suboptimal (eye position >0.9˚ from the fixation point) or the response buttons had been pressed prior to the presentation of the target (i.e., reaction times <10 ms).

**Analysis of event-related potentials (ERPs).** The EEG data was imported and processed offline using EEGLAB [62], an open-source toolbox running under the MATLAB environment. First, an automatic bad channel detection and removal algorithm was applied (using EEGLAB's pop_rejchan method; threshold = 5, method = kurtosis). Later, data of each participant was band-pass filtered with 0.1 Hz as the high-pass cutoff and 40 Hz as the low-pass cutoff frequencies. After this, the epochs were extracted by using a stimulus-locked window of 1900 ms (-700 to 1200 ms) and subjected to an independent component analysis (ICA) algorithm [62]. Blinks and eye-movement artifacts were automatically identified and corrected using an ICA-based automatic method, implemented in the ADJUST plugin of EEGLAB [63]. Bad channels were interpolated using the default spherical interpolation method of the EEGLAB toolbox. Next, data were re-referenced to the average reference. To calculate ERPs, shorter epochs of 900 ms were extracted between -100 ms to 800 ms relative to the onset of the target, and baseline-corrected using the pre-stimulus time interval (-100 to 0 ms).

Our pre-registered ERP analysis focused mainly on the latencies and amplitudes of P1 and N1 components of the stimulus-evoked activity in occipital and parietal visual areas. To this end, ERPs were averaged across a region of interest (ROI) comprising four posterior electrodes (PO7/PO8, O1/O2). The peaks of P1 and N1 components were defined as the most positive (P1) or the most negative (N1) deflections of the grand-average ERP waveforms occurring 70–170 ms or 180–250 ms after the target onset, respectively. P1 and N1 amplitudes were calculated as the mean activity in a 30 ms window centered on each component's respective peak and the timing of the peak defined each component's peak latency. To determine the onset latency of intra- and cross-modal reward modulations, we calculated the difference between high- and low-value ERPs of each cue type within a 10 ms moving window against their baseline difference—within 100 ms before the stimulus onset- [64, 65]. Amplitude differences between 50–800 ms after stimulus onset that reached significance (uncorrected $p < 0.05$) in two or more consecutive time windows are reported [66]. Additionally, we planned to measure the amplitude of P3 component in midline electrodes (Fz, FCz, Cz, CPz, and Pz; 300–600 ms).

After data acquisition and visual inspection of the ERPs, we undertook the following exploratory analyses. Firstly, we performed an exploratory analysis in a time window between 90–120 ms after the stimulus onset (referred to as the PA component). The timing of this component corresponds to the timing of the earliest positive peak of visual ERPs observed in previous studies [67, 68]. As visual areas are known to have a higher sensitivity to the contralateral stimuli [69], we also tested the responses of the posterior ROI to contralateral stimuli (i.e., responses of O1 and PO7 to the stimuli on the right visual hemifield and O2 and PO8 to the stimuli on the left visual hemifield) and measured the correlation of contralateral responses with behavior (see **S5 and S7 Figs**). Lastly, P3 responses (300–600 ms) were also inspected in the posterior ROI (O1, O2, PO7, PO8) in addition to the midline electrodes as visual inspection suggested a more posterior topography at this time interval than expected [70].

**Statistical analysis.** We used 2 by 2 repeated measures ANOVAs (RM ANOVAs) to test the effect of *Reward Value* (high, low) and *Modality* (visual, auditory) on behavioral (d' and RT) and electrophysiological responses during the associative reward learning and orientation discrimination tasks. Planned, paired t-tests were used for pairwise comparisons. Effect sizes in RM ANOVAs are reported as partial eta-squared ($\eta_p^2$) and in pairwise comparisons as Cohen's d; i.e. $d_z$ [71]. Before applying the RM ANOVAs, the assumption of normality was confirmed by inspecting the histograms and Quantile-Quantile (Q-Q) plots of the data. We only observed small deviations from normality in some cases and therefore decided to proceed with our preregistered analysis plan.

To remove the effect of perceptual biases that participants may have for different colors or tone pitches prior to the learning of cues' reward associations, we corrected the behavioral and ERP results of each participant during the orientation discrimination task by subtracting the data of each condition in pre-conditioning from the data in post-conditioning. We note that this method differs from our pre-registered plan where we intended to test the pre-conditioning data separately and rule out significant differences between reward conditions *prior* to learning of reward associations. While our pre-registered plan was suited to test the "statistical significance" of such biases, it did not entirely remove their potential contribution to the effects that occurred *after* learning of reward associations. We therefore decided to employ a stricter correction for such biases and test post-conditioning results after subtraction of pre-conditioning data. The data of the conditioning phase is reported without such correction as the corresponding task was never performed in the absence of reward feedbacks.

## Results

### Conditioning phase

**Behavioral results.** Participants' performance in the localization task employed during the conditioning was near perfect as accuracies for both cues were > 95% (99% ± 1% for auditory and 100%±0% visual cues). Analysis of reaction times (RTs) revealed a significant main effect of modality ($F_{(1,35)} = 70.44$, P < 0.001, $\eta_p^2 = 0.67$). Participants localized visual cues (mean ± s.e.m.: 501 ± 12 ms) faster than auditory cues (mean ± s.e.m.: 584 ± 16 ms), in line with the superior performance of vision in localization tasks [72, 73]. We did not observe a main effect of reward value ($F_{(1,35)} < 1$); however, an interaction was found between reward value and modality ($F_{(1,35)} = 7.68$, p = 0.009, $\eta_p^2 = 0.18$). Post-hoc pairwise comparisons did not reveal a significant effect in either of the modalities: for auditory cues, high reward value slowed down responses (mean ± s.e.m.: 589 ± 17 ms and 580 ± 16 ms for high and low value cues, respectively; t(35) = 1.52, p = 0.137, dz = 0.253), while in the visual modality, high-value cues sped up responses (mean ± s.e.m.: 496 ± 13 ms and 506 ± 12 ms for high- and low-value cues, respectively; t(35) = -2.02, p = 0.051, $d_z = 0.34$).

**ERP responses: P1, N1, P3.** We next examined whether the posterior ROI (PO7/PO8 and O1/O2) that we had planned to use for probing reward effects during the visual orientation discrimination task exhibits reliable reward modulations (ERP amplitude and or latency) during the reward associative learning (**Fig 2** and **Table 1**, see also **S1 Fig**).

**P1 amplitudes** were impacted by modality ($F_{(1,35)} = 12.89$, p < 0.001, $\eta_p^2 = 0.27$), with higher amplitudes for auditory (mean ± s.e.m: 3.66 ± 0.28 μV) than visual cues (mean ± s.e.m: 2.27 ± 0.24 μV). Effects of reward values ($F_{(1,35)} < 1$, $p = 0.919$) and modality by reward interaction ($F_{(1,35)} < 1$, $p = 0.971$) did not reach significance. Analysis of **P1 latency** revealed a main effect of modality ($F_{(1,35)} = 53.1$, $p < 0.001$, $\eta_p^2 = 0.60$), reflecting an earlier P1 peak for auditory (mean ± s.e.m.: 130.7 ± 2 ms) compared to visual cues (mean ± s.e.m.: 155.3 ± 2.7 ms). Neither a main effect of reward value ($F_{(1,35)} = 1.19$, $p = 0.284$) nor an interaction between reward value and modality ($F_{(1,35)} = 2.72$, $p = 0.108$) was found.

Analysis of **N1 amplitudes** in the conditioning phase revealed a main effect of modality ($F_{(1,35)} = 17.42$, p < 0.001, $\eta_p^2 = 0.33$), again with larger amplitudes for auditory (mean ± s.e.m: -1.36±0.35 μV) compared to visual cues (mean ± s.e.m: 0.73±0.32 μV). Neither the main effect of reward value nor the reward by modality interaction reached significance (all ps > 0.1). **N1 latency** was not modulated by modality ($F_{(1,35)} = 1.92$, $p = 0.175$) or by reward value ($F_{(1,35)} < 1$, $p = 0.477$). However, a significant interaction was found between reward value and modality ($F_{(1,35)} = 7.07$, p = 0.012, $\eta_p^2 = 0.17$), reflecting the different directions of value-driven modulation of N1 latencies in the two modalities (**Fig 2B**). Whereas visual high value cues significantly sped up the N1 peak (mean ± s.e.m.: 208.47 ± 2.86 ms and 215.58 ± 2.99 ms and for high and low value cues respectively, t(35) = -2.54, $p = 0.016$, $d_z = 0.424$), auditory high-value cues slightly slowed down the N1 responses, an effect that did not reach statistical significance (mean ± s.e.m.: 218 ± 3.3 ms and 214.7 ± 3.2 ms for high and low value cues respectively, t(35) = 1.12, p = 0.27, Cohen's d = 0.19).

Analysis of the **P3 amplitude** in the posterior ROI (PO7, O1, O2, PO8), quantified between 300 and 600 ms, revealed a main effect of reward value ($F_{(1,35)} = 4.89$, $p = 0.034$, $\eta_p^2 = 0.123$, **Fig 2C**), reflecting larger amplitudes for high-reward cues (Mean ± s.e.m.: 2.87 ± 0.29 μV) than for low-reward cues (Mean ± s.e.m.: 2.65 ± 0.3 μV). The main effect of modality ($F_{(1,35)} < 1$, $p = 0.816$) and an interaction effect between modality and value ($F_{(1,35)} < 1$, $p = 0.458$) were non-significant.

Since learning of reward associations may take time, and behavioral and ERP effects of reward may arise only after associative learning has been completed, we wondered whether

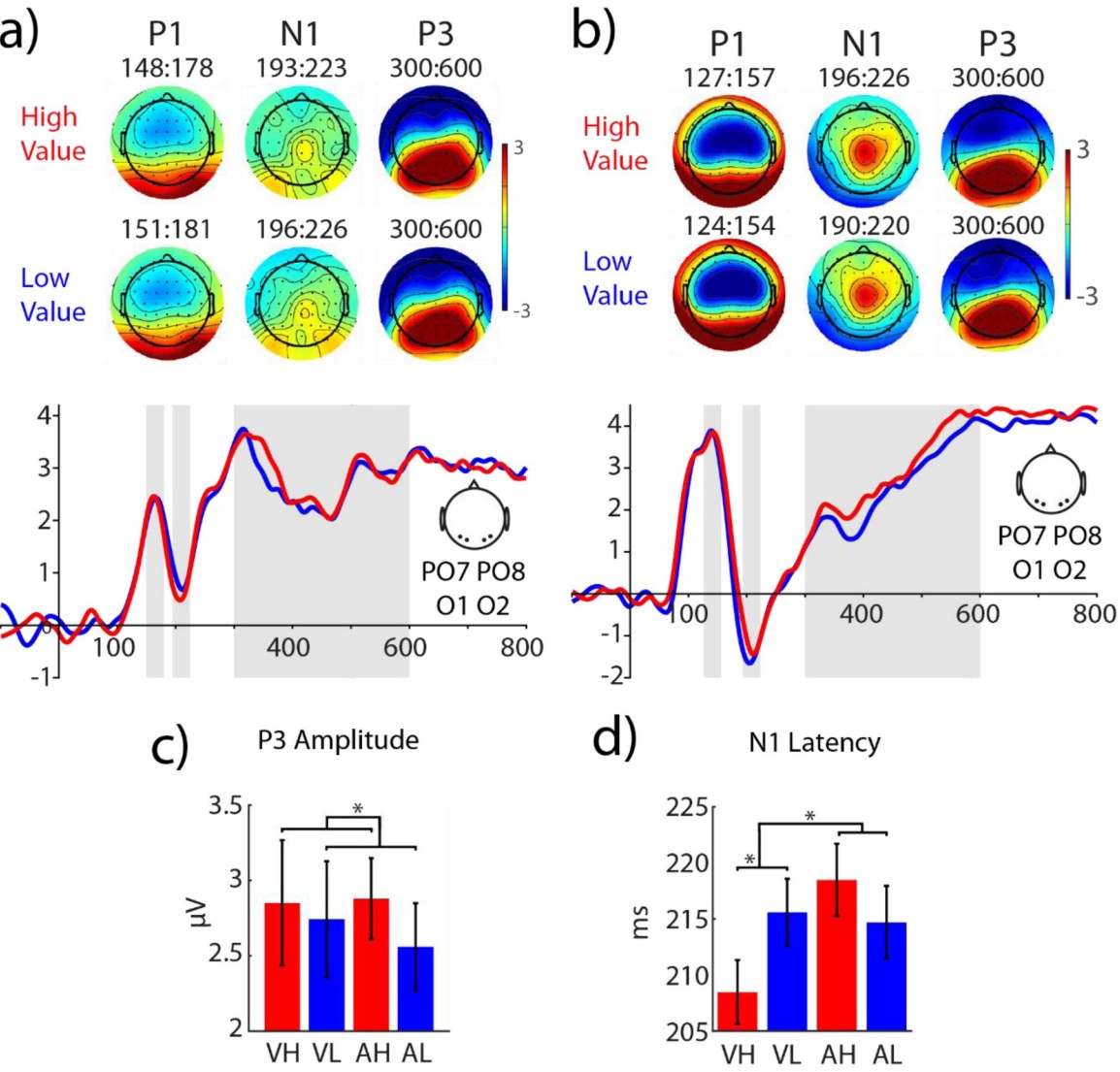

**Fig 2. ERP responses of the posterior ROI to the visual and auditory reward cues during the conditioning phase. a)** ERPs of visual reward cues with high (red traces) and low (blue traces) values measured in a posterior ROI (O1, O2, PO7 and PO8). The shaded grey areas correspond to 30 ms windows around the peak of P1 (70–170 ms) and N1 (180–250 ms) used to estimate the amplitude of these components for each condition (here the window is averaged across high and low value conditions) and the window used to estimate P3 responses (300–600 ms). The topographic distribution of P1, N1 and P3 components are shown for each reward value condition. **b)** Same as **a** for auditory reward cues. **c-d)** analysis of the amplitude and latency of the ERP components revealed significant value-driven modulations for the amplitude of P3 shown in **c** and the latency of the N1 component shown in **d**. Visual High Value: VH; visual Low Value: VL; Auditory High Value: AH; Auditory Low Value: AL.

our reported results differed between the first and the second halves of the conditioning. To test this possibility, we divided the trials for each condition to two halves and entered an additional factor, i.e., phase (first or second half of conditioning), in all of our ANOVAs. We found similar effect sizes in each case (the interaction between reward and modality for RT: $F_{(1,35)}$ = 7.70, p = 0.009, $\eta_p^2$ = 0.18 and for N1 latency: $F_{(1,35)}$ = 7.26, p = 0.011, $\eta_p^2$ = 0.172, and the main effect of reward on P3 amplitude: $F_{(1,35)}$ = 4.27, p = 0.046, $\eta_p^2$ = 0.109), but we did not observe a significant interaction with the phase (all ps>0.1). Therefore, our reported results did not show a dependence on the phase of conditioning, probably because full learning of the reward associations was achieved very fast.

**Table 1. Amplitude and latencies of visual ERP components evoked by visual high value (VH), visual low value (VL), auditory high value (AH), and auditory low value (LA) cues during the conditioning phase, measured in the posterior ROI.**

| Amplitude (µV) | | P1 | N1 | P3 |
|---|---|---|---|---|
| | VH | 2.26 ± 0.25 | 0.62 ± 0.33 | 2.85 ± 0.42 |
| | VL | 2.27 ± 0.26 | 0.84 ± 0.33 | 2.74 ± 0.38 |
| | AH | 3.65 ± 0.29 | -1.22 ± 0.36 | 2.88 ± 0.27 |
| | AL | 3.66 ± 0.31 | -1.49 ± 0.39 | 2.56 ± 0.29 |
| Latency (ms) | | P1 | N1 | |
| | VH | 154.67 ± 3.03 | 208.47 ± 2.86 | |
| | VL | 155.86 ± 3.68 | 215.58 ± 2.99 | |
| | AH | 134.94 ± 3.30 | 218.47 ± 3.23 | |
| | AL | 126.56 ± 2.97 | 214.69 ± 3.24 | |

In summary, examination of the posterior ROI revealed a significant interaction of reward value and reward modality on the latency of N1 responses, with a significant speeding up of ERP responses to high-value compared to low-value visual cues. In the P3 time window, a main effect of reward value was found across all high- compared to low-reward value conditions.

### Post-conditioning phase

**Behavioral results.** To assess the behavioral visual sensitivity, participants' d-prime scores (d' Post *minus* d' Pre) were subjected to an ANOVA with reward value (high or low) and modality (intra- or cross-modal) as independent factors (Table 2 and **Fig 3**). This analysis revealed no main effect of modality ($F_{(1,35)} < 1$, $p = 0.99$) or reward value ($F_{(1,35)} < 1$, $p = 0.35$). Importantly, an interaction was found between modality and reward value ($F_{(1,35)} = 5.75$, p = .022, $\eta_p^2 = 0.14$, **Fig 3E**). Whereas cross-modal reward value significantly enhanced the visual sensitivity (mean ± s.e.m: 0.44 ± 0.19, for the difference between high and low value stimuli, corrected for pre-conditioning, t(35) = 2.31, p = 0.027, $d_z = 0.38$, **Fig 3E**), intra-modal reward value produced an insignificant decrement in visual sensitivity (mean ± s.e.m.: -0.18 ± 0.19, for the difference between high and low value stimuli, corrected for pre-conditioning, t(35) = -0.97, p = 0.34, $d_z = 0.162$). Analysis of the reaction times (RT) revealed an overall slowing down of responses for high- compared to low-value stimuli, but this effect did not reach statistical significance ($F_{(1,35)} = 2.36$, $p = 0.133$). Other main and interaction effects were non-significant (all Fs < 1, all ps > 0.1).

**Table 2. Summary of behavioral and ERP data during the pre- and post-conditioning phases (the gray and white cells, respectively) for intra-modal (IH and IL, high and low values), cross-modal (CH and CI, high and low value), and neutral (Neut) conditions.** Significant pairwise comparisons are shown in bold fonts (p < 0.05).

| | Pre-conditioning | | | | | Post-conditioning | | | | |
|---|---|---|---|---|---|---|---|---|---|---|
| | IH | IL | CH | CL | Neut | IH | IL | CH | CL | Neut |
| RT (ms) | 950 ±25.3 | 968 ±28.2 | 955 ±28 | 961 ±27.8 | 955 ±29.7 | 923 ±21.7 | 917 ±23.4 | 930 ±22.6 | 922 ±20.9 | 906 ±22.8 |
| d' | 2.18 ±0.17 | 2.18 ±0.18 | 1.94 ±0.16 | 2.24 ±0.21 | 2.05 ±0.16 | 1.89 ±0.13 | 2.06 ±0.14 | **1.96 ±0.16** | **1.82 ±0.14** | 1.81 ±0.14 |
| PA (µV) | 0.62 ±0.29 | 0.16 ±0.26 | 3.52 ±0.40 | 3.89 ±0.41 | 0.09 ±0.24 | **0.09 ±0.15** | **0.39 ±0.17** | 3.36 ±0.34 | 3.30 ±0.36 | 0.40 ±0.17 |
| P1 (µV) | 2.28 ±0.36 | 2.35 ±0.35 | 5.56 ±0.43 | 6.03 ±0.39 | 1.98 ±0.34 | 1.84 ±0.26 | 2.07 ±0.30 | 4.81 ±0.34 | 5.00 ±0.36 | 1.93 ±0.26 |
| N1 (µV) | 0.93 ±0.41 | 0.99 ±0.39 | -1.29 ±0.46 | -0.67 ±0.48 | 0.42 ±0.42 | 0.38 ±0.35 | 0.74 ±0.35 | **-0.70 ±0.40** | **-0.94 ±0.41** | 0.91 ±0.36 |
| P3 (µV) | 2.65 ±0.41 | 2.70 ±0.43 | 3.44 ±0.47 | 4.23 ±0.49 | 1.98 ±0.46 | 2.58 ±0.46 | 2.48 ±0.45 | **3.91 ±0.47** | **3.76 ±0.49** | 2.52 ±0.45 |
| P1 (ms) | 151 ±3.63 | 152 ±3.78 | 141 ±3.08 | 145 ±2.78 | 147 ±4.15 | 147 ±4.01 | 147 ±4.33 | 140 ±3.62 | 136 ±3.50 | 151 ±3.26 |
| N1 (ms) | 214 ±3.85 | 213 ±3.71 | 214 ±2.82 | 213 ±2.81 | 211 ±3.59 | 209 ±3.29 | 215 ±3.34 | 213 ±2.66 | 210 ±2.33 | 212 ±3.65 |

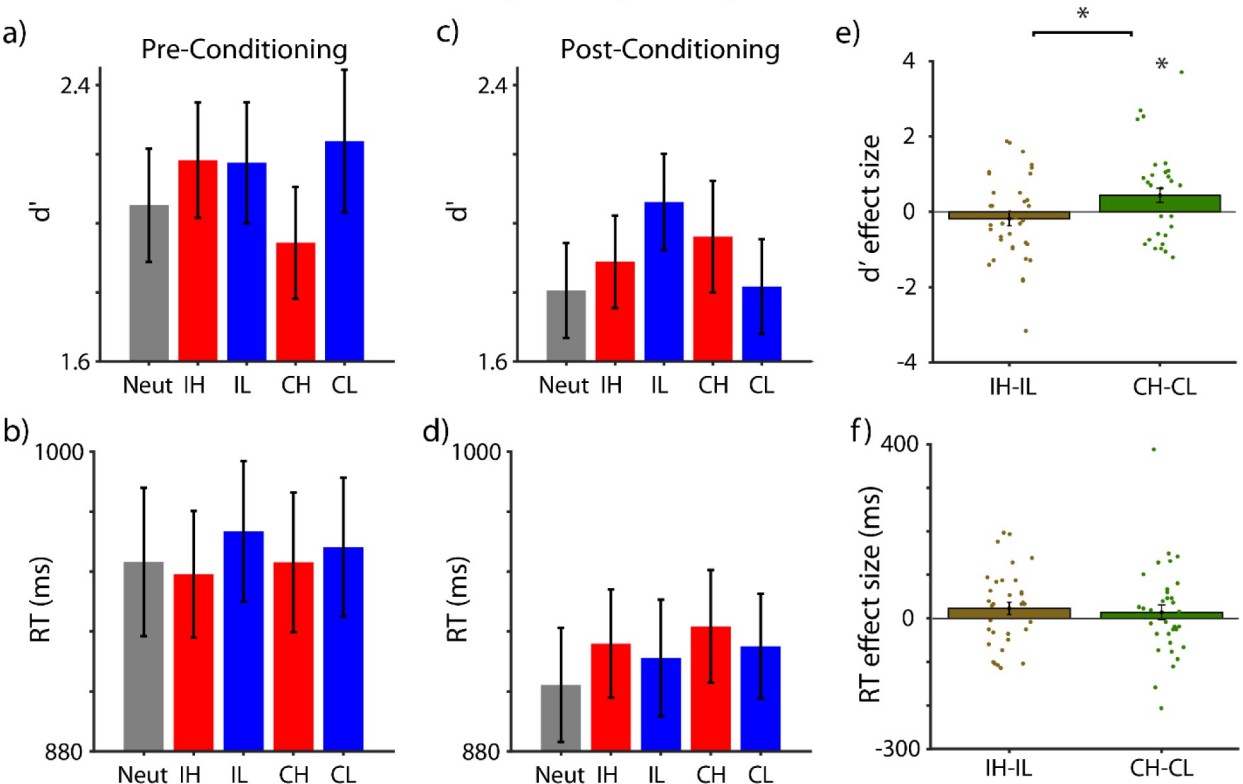

**Fig 3. Behavioral performance in the orientation discrimination task. a)** Visual sensitivity (d-prime: d') during the pre-conditioning phase for different conditions (Neutral: Neut; Intra-modal High Value: IH; Intra-Modal Low Value: IL; Cross-modal High Value: CH; and Cross-Modal Low Value: CL). **b)** Same as **a**, for reaction times (RT). **c)** Visual sensitivity during the post-conditioning phase, i.e. after the associative reward value of different cues were learned. **d)** Same as **c** for RTs. **e)** Effect size for d' modulations, measured as the difference in d' between high- and low-value conditions corrected for their difference during pre-conditioning, in intra- and cross-modal cue types (grey and white bars, respectively). Each dot represents the effect size for one individual subject. **f)** Same as **e** for reaction times (RT). * marks the significant effects (p < 0.05). Error bars are s.e.m.

In summary, during the post-conditioning phase, only cross-modal high-value cues significantly improved the visual sensitivity of orientation discrimination. Reaction times were not significantly affected by the reward value of either type.

**ERP responses in the posterior ROI: Pre-registered analyses.** Examination of ***P1 amplitudes*** with a two-way RM ANOVA revealed a main effect of modality ($F_{(1,35)} = 5.65$, p = 0.023, $\eta_p^2 = 0.14$), corresponding to larger P1 amplitudes in the cross-modal condition than in the intra-modal condition (see **Table 2**). The main effect of reward value ($F_{(1,35)} < 1$, $p = 0.831$) and the interaction between reward value and modality did not reach significance ($F_{(1,35)} < 1$, $p = 0.501$). Analysis of ***P1 latencies*** did not reveal any effect of modality ($F_{(1,35)} < 1$, $p = 0.935$), value ($F_{(1,35)} < 1$, $p = 0.342$), or their interaction ($F_{(1,35)} = 1.2$, $p = 0.287$).

Analysis of the ***N1 amplitude*** revealed a trend for an effect of modality ($F_{(1,35)} = 3.6$, p = 0.066, $\eta_p^2 = 0.093$), with intra-modal reward cues evoking less negative N1 peak than cross-modal reward cues (see **Table 2**). The main effect of reward value did not reach significance ($F_{(1,35)} = 2.04$, p = 0.162) but a trend was found for an interaction between reward value and modality ($F_{(1,35)} = 4.09$, p = 0.051, $\eta_p^2 = 0.105$), where high value cross-modal cues decreased N1 negativity compared to intra-modal condition (see **Table 2**, **Figs 4 and 5** and **S4 Fig**). Planned, pairwise comparisons showed that the cross-modal high value condition significantly decreased the N1 negativity compared to the low value condition (t(35) = 2.98,

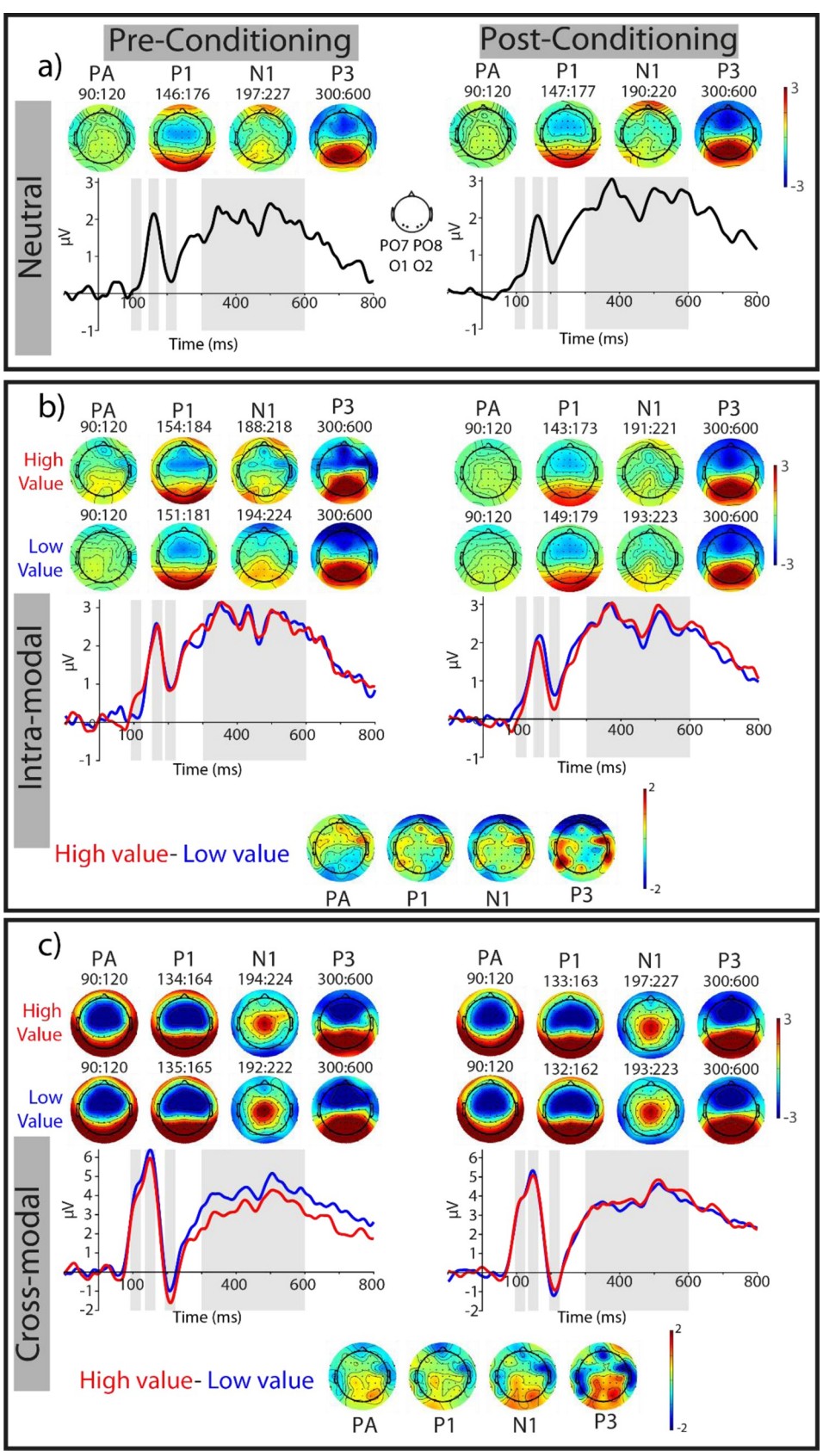

**Fig 4. ERP responses of the posterior ROI (PO7, PO8, O1, O2) during the visual orientation discrimination task.** **a)** ERPs elicited by the neutral condition during pre- and post-conditioning (illustrated on the left and right, respectively). The topographic distributions of PA (90–120 ms), P1 (most positive peak 70–170 ms), N1 (most negative peak 180–250 ms), and P300 (300–600 ms) components, measured in respective grey shaded areas of the ERP time courses are also shown. **b)** ERPs in the presence of task-irrelevant, intra-modal reward cues, with high (red traces) and low (blue traces) values. The corresponding topographic distribution of each component is shown for each reward value condition. To test the significance of value-driven modulations, the difference between high- and low-value conditions was corrected for their pre-conditioning difference. The topographic distributions of these corrected modulations are shown for each component (the lowermost topographic maps). See also Fig 5C and **S4 Fig**. **c)** Same as **b** for ERPs in the presence of task-irrelevant, cross-modal reward cues.

p = 0.005, $d_z$ = 0.5, **Fig 5C**). The difference between intra-modal, high and low value conditions did not reach statistical significance (p = 0.45). **Latency analysis of N1 component did not reveal any effects of modality** ($F_{(1,35)} < 1$, $p = 0.935$), value ($F_{(1,35)} < 1$, $p = 0.363$) or their interaction ($F_{(1,35)} = 2.17$, $p = 0.150$).

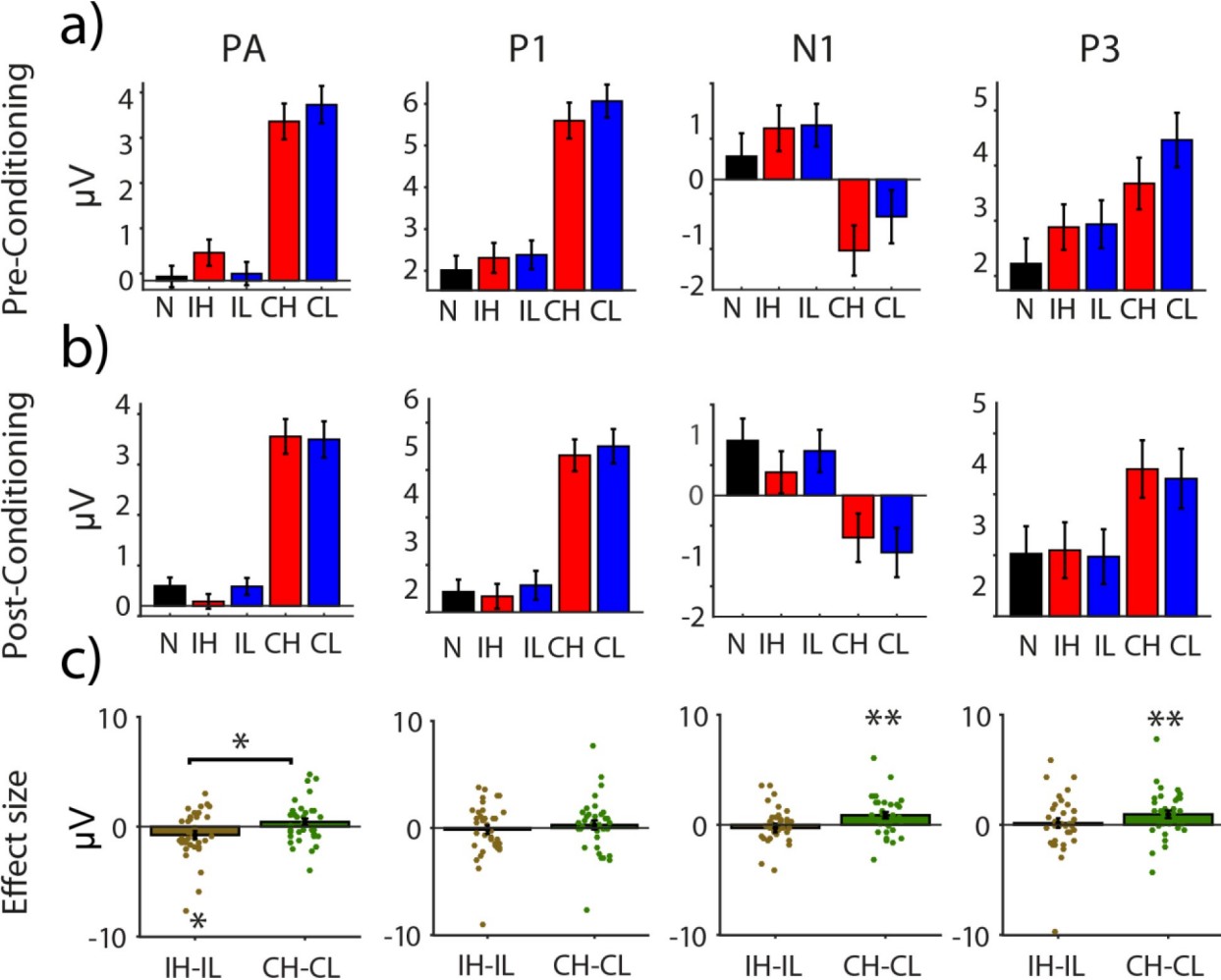

**Fig 5. Mean amplitude of different ERP components of the posterior ROI (PO7, PO8, O1, O2) during the visual orientation discrimination task.** **a)** mean amplitude of PA, P1, N1 and P3 responses evoked by neutral (N), intra-modal high value (IH), intra-modal low value (IL), cross-modal high value (CH) and cross-modal low value (CL) conditions, during pre-conditioning. **b)** Same as **a**, for ERPs during post-condoning. **c)** Corrected effect sizes, measured as the difference between high- and low-value conditions corrected for their difference during pre-conditioning, for intra- and cross-modal cue types (grey and white bars, respectively). * mark significant differences (p< 0.05) and ** mark significant differences (p< 0.01). Each dot represents the effect size for one individual subject. Error bars are s.e.m.

Based on our pre-registered analysis plan, we next examined the earliest time points where the reward modulations of each cue type reached significance using a moving window analysis (see Methods). This analysis revealed an earlier onset of value-driven modulations for intra-modal compared to cross-modal conditions in the posterior ROI. High- and low-value ERP responses of the intra-modal conditions differed significantly within the P1 window, i.e. between 101 and 114 ms. The cross-modal value-modulations started later, at 167 ms, and remained significant until 223 ms, thus overlapping with the N1 time window (Fig 4).

**P3 responses at midline electrodes.**   Analysis of the P3 component in midline electrodes (300–600 ms, corrected for pre-conditioning) revealed no main effect of modality, reward value, electrode, or their interaction (all ps > 0.1). Although inspection of the ERP responses of midline electrodes (see also **S2 and S3 Figs**) suggests that value-driven modulations occur in other time windows than our pre-registered intervals, we decided to adhere to our a priori plan and do not explore these modulations any further.

**ERP responses in the posterior ROI: Exploratory analyses.**   Based on the timing of the earliest positive peak of visual ERPs (i.e. 90–120 ms) observed in classical studies of visual perception [67, 68], we next performed an exploratory analysis on the amplitude of evoked responses in this time window (referred to as the PA component throughout, see **Figs 4 and 5** and **S4 Fig**). A two-way ANOVA on PA amplitudes revealed a significant value × modality interaction effect ($F_{(1,35)} = 5.12$, p = 0.03, $\eta_p^2 = 0.128$), but no main effect of reward value or modality (both Fs < 1 and ps > 0.1). Post-hoc pairwise comparisons (**Fig 5C**) revealed a significant suppression of PA amplitude for high- compared to low-value intra-modal condition (t(35) = -2.1, p = 0.04, $d_z = 0.35$). The value-driven modulation in the cross-modal conditions was in the opposite direction but did not reach statistical significance (t(35) = 1.35, p = 0.185, $d_z = 0.22$). We wondered whether the negative modulation of PA responses by intra-modal reward value reflects a suppression of high-reward stimuli or rather an enhancement of the low-reward conditions. To this end, we contrasted the PA amplitudes of intra-modal high- and low-value conditions, respectively, against the neutral condition that had not undergone reward associative learning (see also **Figs 4 and 5**). These comparisons revealed a significant suppression of PA responses evoked by intra-modal, high-value compared to neutral stimuli (t(35) = -2.07, p = 0.04, $d_z = 0.35$). The intra-modal, low-value stimuli, however, were not significantly different from the neutral condition (t(35) = 0.25, p = 0.80, $d_z = 0.042$). These results suggest that intra-modal high-value stimuli undergo active suppression compared to both intra-modal low value and neutral conditions occurring very early after the onset of stimuli.

We next inspected the ERPs of the posterior ROI in a later time window (300–600 ms) corresponding to the *P3 component* (see **Figs 4 and 5** and **S4 Fig**). This analysis revealed a main effect of reward value ($F_{(1,35)} = 5.63$, *p* = 0.023, $\eta_p^2 = 0.138$). The main effect of modality and the modality by reward interaction were not significant (both ps > 0.1). Post-hoc pairwise comparisons revealed a significant enhancement of P3 responses by cross-modal high- compared to low-value condition (t (35) = 2.72, P = 0.01, d = 0.45). In the intra-modal condition, P3 amplitudes did not differ between the two reward conditions (t(35) = 0.37, *p* = 0.72, $d_z = 0.061$). Thus, examination of late ERPs in the posterior ROI overall indicated a value-driven response enhancement for the cross-modal condition (**Fig 5C**).

Taken together, the ERP effects provide evidence for an interaction effect between intra- and cross-modal reward cues, with intra-modal cues leading to an early suppression and cross-modal reward cues producing a later response enhancement for high- compared to low-value stimuli. Importantly, we excluded the possibility that these effects were driven by changes in eye position in response to different reward conditions (see **S1 Text**). Similar results were obtained when contralateral ERPs were examined or when only correct trials were included in our analysis (see also **S1 Text**, **S5 and S6 Figs**).

**Correlation of behavioral and ERP amplitudes.** We next measured the correlation between value-driven modulations of ERP amplitudes for which a significant reward effect was found (i.e., PA, N1 and P3, see **Fig 5**) and behavioral indices (d' and RT). This pre-registered analysis did not reveal any significant correlation (all ps> 0.1). However, an exploratory analysis revealed a significant positive correlation between value-driven modulations of the contralateral N1 and P3 components and behavioral d-primes in cross-modal stimuli (see **S7 Fig**).

## Discussion

In the current study we tested whether the perception of a compound stimulus comprising a visual target and task-irrelevant intra-modal or cross-modal cues is affected by the reward associations of task-irrelevant components. To this end, we examined behavioral and electrophysiological responses to reward cues during a conditioning phase when reward associations were learned and during a post-conditioning phase when rewards were not delivered anymore and reward cues were irrelevant to the visual discrimination task that participants had to perform. In the conditioning phase, we found that intra- and cross-modal reward cues affect latency of the N1 responses over the posterior electrodes differently: while intra-modal reward cues significantly sped up N1 response, cross-modal reward cues only led to an insignificant deceleration of N1 responses. However, an exploratory analysis of P3-like responses of the posterior ROI revealed higher amplitudes for both intra- and cross-modal high value cues. In the post-conditioning phase, similarly to conditioning phase, P3-like responses of the posterior ROI were enhanced for high value cues of both types. Importantly, the effect of intra- and cross-modal rewards on earlier components of posterior ERPs differed. Cross-modal stimuli led to a later reward modulation at a time point that corresponded to the N1 component, whereas intra-modal reward modulations occurred earlier and an exploratory analysis revealed that high reward cues significantly suppressed posterior ERPs relative to low reward cues within 90–120 ms (PA window). Behavioral results were in line with electrophysiological findings; cross-modal reward cues significantly enhanced visual sensitivity whereas intra-modal reward cues led to a weak suppression. Interaction effects of reward value with modality within PA and N1 windows suggest that intra-modal and cross-modal reward values exert different effects on perception of a compound object under settings employed in our study (i.e., task-irrelevant reward cues during a no-reward phase). Similar reward effects of intra- and cross-modal conditions in the P3 window, observed in our exploratory analysis, indicate that beyond the differential effect of modality on the early components of posterior ERPs, the later reward effects are independent of the sensory modality.

### Reward-driven modulations during conditioning

In this phase, a visual target predictive of higher rewards sped up reaction times and early cue-evoked neural responses, particularly in the N1 window. This result is similar to early modulations of visual ERPs (i.e. <250 ms) observed in previous studies [15, 17, 54, 70, 74, 75]. Auditory stimuli also evoked strong responses in the posterior ROI even in the absence of concomitant visual stimulation [76–79]. However, reward signals from auditory stimuli did not significantly affect the amplitude or latency of early ERP components when auditory stimuli were presented alone. Lack of cross-modal reward effects during the conditioning phase may indicate that reward information does not transfer automatically when there is no incoming visual information.

The similarity of intra- and cross-modal reward effects on later P3 component, observed in an exploratory analysis, indicates that at the later stages of sensory processing reward

information integrates into a coherent reward representation across various sources irrespective of the unique characteristics of those sources. On the other hand, during the early stages of sensory processing, privileged processing of rewarded stimuli only prioritizes cues that are most suited for the task at hand, i.e., vision in case of a task requiring localization. These results are overall explainable within the framework of reinforcement learning, where association of a stimulus with reward results in value-driven modulations at 2 time points [74]: an early modulation of neural responses for stimuli associated with higher value within the first 200–250 ms after the stimulus onset [54, 74]; and a later reward modulation primarily the P3 component related to anticipation of the reward delivery [17, 74, 80–82].

### Reward-driven modulations during post-conditioning

The results obtained in the post-conditioning phase showed a differential pattern of reward-related modulation for cross-modal and intra-modal cues. Specifically, we observed an improvement of behavioral measures of visual sensitivity for high compared to low-value cross-modal cues. These results are in line with the reported facilitatory effects of cross-modal value on visual processing observed in human psychophysical and neuroimaging studies [21, 61]. As a first attempt to characterize the electrophysiological correlates of this effect, our study demonstrated that the behavioral advantage conferred by high-reward auditory cues is accompanied by a modulation of posterior ERPs. As these modulations were corrected for differences potentially occurring due to physical stimulus features (i.e., tone pitches) already during the pre-conditioning phase, they most probably reflect effects driven by the associative learning of the reward values. However, we note that overall, the effect sizes in our study were small, likely due to the fact that reward cues were not the task-relevant features of the stimulus and did not predict the delivery of reward anymore, factors that can be further investigated in future studies.

Cross-modal reward modulations in our study occurred later than attentional effects observed in some of the previous studies of cross-modal attention [83], where response modulations were found in P1 window. However, in these experiments auditory tones were presented prior to the visual target and could therefore modulate the early ERP responses. In fact, when auditory and visual components of an audiovisual object were presented together, attentional effects occurred at a later time window around 220 ms [84, 85]. Whereas reward-driven boost of attention may to some extent account for cross-modal reward effects in our study, the direction of response modulations suggests that additional mechanisms may also be involved (**Fig 5C**). Specifically, we found a reduction in N1 negativity after learning of the reward associations, which is different from an enhanced P1 positivity and N1 negativity that has been observed in studies of cross-modal attention [83–85]. One possible mechanism is a reward-driven enhancement of audiovisual integration, which is in line with recent findings demonstrating a role of reward in multisensory integration [44, 45]. This mechanism can also explain the direction of ERP modulations, as previous studies found that an audiovisual stimulus elicits response modulations of visual ERPs mainly in N185 window, with a reduction of the negativity of N185 component compared to the unimodal stimuli [64], which is similar to the pattern of modulations we observed for cross-modal rewards. The reduction in N1 negativity may indicate that audiovisual integration enhances the gain of the neural responses of visual cortex, hence reducing the energy (i.e. N1 amplitude) needed to process the same load of sensory information, an example of sub-additive cross-modal interactions reported before [48, 86, 87]. An enhanced integration between auditory reward cues and visual target in our task potentially reduces the distracting effect of task-irrelevant sounds on visual discrimination while promoting the spread of privileged processing from rewarded sounds to the visual target. Later modulations by cross-modal compared to intra-modal rewards suggests that a putative

reward-driven enhancement of audiovisual integration may first occur in multimodal areas such as in Superior Temporal Sulcus (STS), being then fed back to visual cortex, a proposal in line with the findings of neuroimaging studies of cross-modal value and emotion effects on vision [21, 88]. In addition to the above mechanisms (reward-driven boost of attention and audiovisual integration), the later value-driven modulations in P3 window found in an exploratory analysis may indicate that the cross-modal reward effects also rely on post-sensory and decisional stages [89].

The effects of intra-modal reward cues on visual perception were contrary to our a-priori hypothesis that intra-modal and cross-modal reward stimuli should have similar facilitatory effects on perception of a compound object. This hypothesis was based on previous findings [11] that reported a spread of reward enhancement effects from one component of an object to its other parts (here from colored circles signaling reward to the Gabor target), akin to the spread of object-based attention [47, 90]. Contrary to our prediction, our exploratory analysis revealed that intra-modal reward stimuli interfered with sensory processing of the visual target, which was reflected in suppressed ERP responses at an early time window of 90–120 ms elicited by high rewards compared to both low reward and neutral conditions. The interference effect that we observed is in line with the findings of several studies where value-driven effects during visual search were investigated [30, 91–99]. These experiments have consistently reported that presentation of a high reward stimulus at the target location speeds up visual search and enhances target-evoked responses, whereas presentation of high reward stimuli at the distractor location captures attention away from the target and interferes with the search task. Given these previous findings, we superimposed reward cues on the target to boost the processing of all object elements at that location. However, since the visual discrimination task was performed on a different feature of the object (i.e., orientation) than the defining feature of the reward cue (color), high reward visual cues may have captured attention away from the target feature, hence interfering with the target processing. A similar interference effect has been observed in studies where a certain feature of an object was predictive of its reward but this feature was incongruent with the goal of the task, e.g. in many tasks employing the Stroop Effect [100–104].

In the light of these previous studies, we propose three possible mechanisms for the observed intra-modal, reward-related suppression. Firstly, it is possible that as a result of an enhanced response to the high-reward, task-irrelevant cues [105] some form of local inhibition is exerted on the adjacent stimuli, thereby decreasing the overall responses to target and rewarding cues that were at the same location. Such a center-surround inhibition around an attended feature has been observed in studies of feature-based and spatial attention [106–108]. Secondly, it is possible that the suppression is a reflection of the higher processing load of high-reward cues [109] and the capacity limitation of attentional processing. This mechanism could therefore result from a mixture of enhanced processing of task-irrelevant reward cues in some trials and decrement of processing of *target+task-irrelevant cues* in other trials due to the depletion of attentional resources. Thirdly, it is possible that the suppression is due to cognitive control mechanisms that actively inhibit the processing of intra-modal reward cues with a resultant spillover of inhibition to the target as also observed in previous studies of feature-based attention [110]. Overall, the above scenarios all indicate that the privileged processing of high reward intra-modal cues (as shown during the conditioning) did not effectively spread to the visual target. Therefore, it is possible that our paradigm failed to promote the integration of task-irrelevant intra-modal reward cues and the target into a coherent object. This issue can be remedied by increasing the duration of training on the task or using more object-like stimuli. Future studies will be needed to tease apart these possibilities.

Taken together, our study demonstrates that the perception of an object can be influenced by the reward value of its constituent components both intra- as well as cross-modally. The

differences that we observed between the intra-modal and cross-modal conditions might be due to the specific features of our experimental design (e.g., task-irrelevant cues that were previously associated with rewards), and some of our reported effects were observed while employing exploratory analyses and hence they await future replications and extensions. In this vein, our results also indicated that the perceived intensity of visual and auditory stimuli might have been different, as both P1 and N1 components were stronger for auditory compared to visual stimuli during the conditioning phase. Since we quantified the ERP measures separately for each sensory modality, calculated the reward effects against stimuli from the same modality, and corrected for pre-conditioning biases, it is unlikely that our reported results are due to the differences in perceived intensity of auditory and visual stimuli. We note however that using an adaptive method to equalize the perceived intensity of auditory and visual stimuli [111] would be an interesting addition to future studies. Despite these limitations, the differential modulations by the intra-modal and cross-modal reward cues may point to distinct neural pathways mediating the effects of reward from the same or different sensory modality on visual perception, a possibility that can be further examined by employing methods with a better spatial resolution. Embedding cross-modal rewards in visual tasks is a promising tool to assist vision, especially in the face of visual impairments, through boosting cross-modal advantages conferred by another intact sensory modality and comparing reward effects across intra-modal and cross-modal cues provides a first critical step towards this aim.

## Supporting information

**S1 Text.**
(DOCX)

**S1 Fig. ERPs of midline electrodes during the conditioning phase.**
(DOCX)

**S2 Fig. ERPs of midline electrodes during the pre-conditioning phase.**
(DOCX)

**S3 Fig. ERPs of midline electrodes during the post-conditioning phase.**
(DOCX)

**S4 Fig. This figure should be compared to Fig 4 in the main text.**
(DOCX)

**S5 Fig. Contralateral responses of the posterior ROI during pre- and post-conditioning phases, see also Fig 4 in the main text.**
(DOCX)

**S6 Fig. ERP results of the posterior ROI when only correct trials were included, see also Fig 4 in the main text.**
(DOCX)

**S7 Fig. Correlation between electrophysiological and behavioral effects of reward value.**
(DOCX)

## Acknowledgments

We thank Adem Saglam for his help with programming of the experiment, Franziska Ehbrecht for her help with the data collection, and Jessica Emily Antono for her valuable comments on the manuscript.

## Author Contributions

**Conceptualization:** Roman Vakhrushev, Arezoo Pooresmaeili.

**Data curation:** Roman Vakhrushev.

**Formal analysis:** Roman Vakhrushev, Felicia Pei-Hsin Cheng, Arezoo Pooresmaeili.

**Funding acquisition:** Arezoo Pooresmaeili.

**Investigation:** Roman Vakhrushev, Arezoo Pooresmaeili.

**Methodology:** Roman Vakhrushev, Felicia Pei-Hsin Cheng, Anne Schacht, Arezoo Pooresmaeili.

**Project administration:** Arezoo Pooresmaeili.

**Resources:** Arezoo Pooresmaeili.

**Software:** Roman Vakhrushev.

**Supervision:** Arezoo Pooresmaeili.

**Validation:** Roman Vakhrushev, Anne Schacht, Arezoo Pooresmaeili.

**Visualization:** Roman Vakhrushev.

**Writing – original draft:** Roman Vakhrushev, Arezoo Pooresmaeili.

**Writing – review & editing:** Roman Vakhrushev, Felicia Pei-Hsin Cheng, Anne Schacht, Arezoo Pooresmaeili.

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
