## [Decision Letter · Decision Letter 0]

19 Dec 2022

PONE-D-22-27111Differential effects of intra-modal and cross-modal reward value on perception: ERP evidencePLOS ONE

Dear Dr. Pooresmaeili,

Thank you for submitting your manuscript to PLOS ONE. After careful consideration, we feel that it has merit but does not fully meet PLOS ONE’s publication criteria as it currently stands. Therefore, we invite you to submit a revised version of the manuscript that addresses the points raised during the review process.

The main points to be reviewed are the clarification of the research design and some substantial modifications required in exposing significant and non-significant effects. In particular, we ask why a visual neutral reward cue was used, which was not done for acoustic rewards. This point is essential to remove any doubts regarding the possible influence of this factor on the results.

Greater clarity is required in distinguishing results that are significant from those that indicate a trend, from those that are statistically insignificant.

Furthermore, some concerns are indicated about the normalization process to remove perceptual biases due to color or tone perception.

We also remind you that PLOS ONE has a specific policy regarding data availability (see https://journals.plos.org/plosone/s/data-availability). We understand that the authors have given the possibility to consult the data on request during the peer-review process, and that all data will be available after acceptance (in the form you report: "the URLs/accession number/DOIs will be available only after acceptance of the manuscript for publication so that we can ensure their inclusion before publication"). We ask for confirmation.

We look forward to receiving your revised manuscript.

Kind regards,

Nicola Megna, M.D.

Academic Editor

PLOS ONE

Journal Requirements:

3. We note that you have stated that you will provide repository information for your data at acceptance. Should your manuscript be accepted for publication, we will hold it until you provide the relevant accession numbers or DOIs necessary to access your data. If you wish to make changes to your Data Availability statement, please describe these changes in your cover letter and we will update your Data Availability statement to reflect the information you provide

4. "Please update your submission to use the PLOS LaTeX template. The template and more information on our requirements for LaTeX submissions can be found at " ext-link-type="uri" xlink:type="simple">http://journals.plos.org/plosone/s/latex."

"We thank Adem Saglam for his help with programming of the experiment, Franziska Ehbrecht for her help with the data collection, and Jessica Emily Antono for her valuable comments on the manuscript. This work was supported by an ERC Starting Grant (no: 716846) to AP. The funders had no role in study design, data collection and analysis, decision to publish, or preparation of the manuscript"

"This work was supported by an ERC Starting Grant (no: 716846) to AP. The funders had no role in study design, data collection and analysis, decision to publish, or preparation of the manuscript."

Reviewers' comments:

Reviewer's Responses to Questions

**Comments to the Author**

1. Is the manuscript technically sound, and do the data support the conclusions?

Reviewer #1: Partly

Reviewer #2: Partly

2. Has the statistical analysis been performed appropriately and rigorously? 

Reviewer #1: No

Reviewer #2: No

3. Have the authors made all data underlying the findings in their manuscript fully available?

Reviewer #1: No

Reviewer #2: No

4. Is the manuscript presented in an intelligible fashion and written in standard English?

Reviewer #1: Yes

Reviewer #2: Yes

5. Review Comments to the Author

Reviewer #1: In this study, the authors investigated the effect of rewarding intra-modal or cross-modal cues on visual orientation discrimination using a combination of psychophysics and EEG. From a behavioral point of view, they found that highly rewarded sounds improved visual discrimination sensitivity whereas highly rewarded visual cues tended to interfere with discrimination performance. EEG data strengthened this dissociation: while highly rewarded visual cues elicited an early suppression of ERPs of posterior electrodes, highly rewarded sound cues produced an enhancement of amplitude of later (from N1 to P3) responses relative to sound cues associated with lower reward. The authors concluded that, while the reward magnitude can modulate sensory processing, it does so differently based on the involved sensory modalities.

This is a very interesting study addressing a novel research question. The paper is also mostly well written. ERP analyses and the corresponding drawn conclusions look reasonable to me, but I do not have expertise with EEG, therefore I hope other reviewers could evaluate more deeply this part of the study and support that it is appropriate. However, I do have some concerns related to other methodological and data analysis aspects which I feel should be addressed before recommending this work for publication.

MAJOR CONCERNS

For clarity reasons, I report my major concerns as three different points. However, I believe they are all interrelated.

- If I correctly understood the study design, the authors have included a neutral visual condition (grey cue) whereas a corresponding neutral auditory condition is missing. I was wondering why the authors have opted for this unbalanced design which may, in my opinion, affect the results in different ways. For instance, the probability of an intramodal cue is higher than the probability of a cross-modal cue which may influence participants’ expectation and, doing so, also the performance in the task.

- The authors analyzed data using RM ANOVAs with Reward value (high, low) as one of the two factors. However, the study design included a neutral condition which has not been compared to the reward conditions. I was wondering why the authors did not include a Condition factor in their analyses with three levels (high reward, low reward, neutral).

- The authors normalized data to remove the effect of perceptual biases that subjects may have for different colors or tone frequencies. This is, in my opinion, a correct procedure since there is evidence that, for instance, RT may be influenced by stimulus color even for isoluminant stimuli. However, the normalization procedure does not look appropriate to me. If I correctly understood, the authors subtracted the data of each condition in pre-conditioning from its counterpart in post-conditioning. This procedure allows to select the effect of reward but it does not eliminate the bias for specific chromaticies or tone frequencies. In my opinion, a possible correct method, given the unbalanced design, would be identifying each subjects bias in pre-conditioning (e.g., the difference between orange and blue in RTs, or the difference between high and low tone frequency) and correct the post-conditioning data by adding or subtracting the difference observed in pre-conditioning to limit if not eliminate the color (or pitch) bias. For instance, if a subject in pre-conditioning is 6 ms faster in the orange compared to the blue cue condition, 6 ms could be added in the orange condition post-conditioning data. In this way, the remaining difference between orange and blue in post-conditioning may be due to different reward value (plus noise). After this correction, the authors could implement the ANOVAs as defined.

MINOR CONCERNS

- The experimental procedures section requires more details. For instance, the authors did not explain what kind of calibration procedure they used and for how long it lasted. Similarly, they did not specify the number of sessions they used in the QUEST method as well as the number of trials. As for the conditioning phase, it is not clear what happened after a wrong response. What kind of feedback did the participants receive in this situation? Finally, did the authors checked for data normality?

- Rows 359-360: please report p-values of t-tests.

- Rows 392-394: the authors stated that auditory high-value cues tended to slow down N1 responses. I think the analysis does not support this statement since the p-value (0.27) is more compatible with a lack of difference. We usually indeed refer to a trend for p-values comprised between 0.05 and 0.10.

- In general (e.g., Table 2), the authors used acronyms such as IH and CL also for the pre-conditioning conditions. This may be confusing for the reader since the cues were not conditioned yet. Perhaps, I would clarify this aspect in the manuscript.

- Figure 3E: what does the double asterisk mean?

- FIGURES 2,3,5: the authors are using a color code for bars which is not defined in the captions.

- The manuscript requires some editing. References are not consistently formatted (e.g., row 73) and there are many language mistakes (e.g., ‘auditory tones’, row 177: ‘different’ instead of ‘difference’) and typos (e.g., caption Figure 5: ‘pre-condoning’; row 645: ‘enchantment’; row 664: ‘at a the’)

Reviewer #2: In this very interesting study, the authors aimed to investigate the effect of modality (acustic or visual) and value of a reward in an orientation discrimination task, using both psychophysical and electrophysiological measures. They found that a cross-modal highly rewarded cue is efficient in improving visual sensitivity, while intra-modal reward cues tend to interfere with a visual task.

There are some concerns about the experimental design and the statistical analysis.

MAJOR CONCERNS:

- The authors comment continuously not significant data, and I think this can be correct when this is underlined. When you read the article you have the impression of solid results, but at the end of many paragraph you discover that we are talking about trends. Even, on line 394, we read of a trend only to discover that the p value is 0.27. This is clearly not a trend. This is particular evident also in lines 420-437. I ask the authors, to make everything clearer, to break down the significant effects identified and talk about them first, and then proceed to identify the trends, and then reformulate everything.

- The value of reward is not well balanced in the visual and auditory modality: there are three values for the visual cues (high-neutral and low value) and only two for the auditory cues (low and high). This is an important issue to me, because the salience of visual cues may be decreased for this very reason. The problem is that the pejorative effect of visual cues could be due to this aspect of the experimental design. Authors should therefore at least explain why they used a neutral condition, why they also didn't present the data in the neutral condition, or better, describe also this condition (neutral visual cue). Naturally, they should convince readers that the pejorative effect of visual cues is not due to this factor.

MINOR CONCERNS:

- There are many confusing aspects: acronyms, such like IH, CH, etc could be confusing. The experimental design should be described with more clarity. Figure 1c doesn't help (I suggest to modify or eliminate it). The pitch function is not immediately clear, because at rows 211-213 the reader has to infer it. line 225: maybe it is "cues", not "stimuli". There are also some typos to be addressed.

6. PLOS authors have the option to publish the peer review history of their article (what does this mean?). If published, this will include your full peer review and any attached files.

Reviewer #1: No

Reviewer #2: **Yes: **Nicola Megna

---

## [Author Response · Author response to Decision Letter 0]

17 Feb 2023

Summary of the decision letter and our response*

[*In the PDF version of our rebuttal letter, the editor’s and reviewers’ comments are in black, our responses are in blue, and the revised sections in the manuscript are marked in green. Please note that the line numbers we refer to in our responses reflect the line numbers in the manuscript with track changes.]

Editor’s email:

PONE-D-22-27111

Differential effects of intra-modal and cross-modal reward value on perception: ERP evidence

PLOS ONE

Dear Dr. Pooresmaeili,

Thank you for submitting your manuscript to PLOS ONE. After careful consideration, we feel that it has merit but does not fully meet PLOS ONE’s publication criteria as it currently stands. Therefore, we invite you to submit a revised version of the manuscript that addresses the points raised during the review process. 

The main points to be reviewed are the clarification of the research design and some substantial modifications required in exposing significant and non-significant effects. In particular, we ask why a visual neutral reward cue was used, which was not done for acoustic rewards. This point is essential to remove any doubts regarding the possible influence of this factor on the results.

Greater clarity is required in distinguishing results that are significant from those that indicate a trend, from those that are statistically insignificant.

Furthermore, some concerns are indicated about the normalization process to remove perceptual biases due to color or tone perception.

We also remind you that PLOS ONE has a specific policy regarding data availability (see https://journals.plos.org/plosone/s/data-availability). We understand that the authors have given the possibility to consult the data on request during the peer-review process, and that all data will be available after acceptance (in the form you report: "the URLs/accession number/DOIs will be available only after acceptance of the manuscript for publication so that we can ensure their inclusion before publication"). We ask for confirmation.

We thank the editor and the reviewers for their positive evaluation of our work and for their insightful suggestions. In our revisions, we have carefully implemented all reviewers’ suggestions, which have made the manuscript much stronger than the original version. We have also uploaded the data and the analysis scripts to the Open Science Framework (OSF), as outlined in the cover letter and the manuscript. We hope that we have satisfactorily addressed all reviewers’ comments and our manuscript can be now accepted for publication in Plos One. 

Point-by-point response

Comments of Reviewer 1

Reviewer #1: In this study, the authors investigated the effect of rewarding intra-modal or cross-modal cues on visual orientation discrimination using a combination of psychophysics and EEG. From a behavioral point of view, they found that highly rewarded sounds improved visual discrimination sensitivity whereas highly rewarded visual cues tended to interfere with discrimination performance. EEG data strengthened this dissociation: while highly rewarded visual cues elicited an early suppression of ERPs of posterior electrodes, highly rewarded sound cues produced an enhancement of amplitude of later (from N1 to P3) responses relative to sound cues associated with lower reward. The authors concluded that, while the reward magnitude can modulate sensory processing, it does so differently based on the involved sensory modalities.

This is a very interesting study addressing a novel research question. The paper is also mostly well written. ERP analyses and the corresponding drawn conclusions look reasonable to me, but I do not have expertise with EEG, therefore I hope other reviewers could evaluate more deeply this part of the study and support that it is appropriate. However, I do have some concerns related to other methodological and data analysis aspects which I feel should be addressed before recommending this work for publication.

We thank the reviewer for the overall positive evaluation of our work and the insightful suggestions which have greatly helped us to improve our manuscript.

MAJOR CONCERNS

For clarity reasons, I report my major concerns as three different points. However, I believe they are all interrelated.

- If I correctly understood the study design, the authors have included a neutral visual condition (grey cue) whereas a corresponding neutral auditory condition is missing. I was wondering why the authors have opted for this unbalanced design which may, in my opinion, affect the results in different ways. For instance, the probability of an intramodal cue is higher than the probability of a cross-modal cue which may influence participants’ expectation and, doing so, also the performance in the task.

We thank the reviewer for raising this point. As outlined below in detail, the neutral condition was never presented during the conditioning and participants did not associate the neutral cue with any reward. This important point is now clarified in lines: 221-224, 238-240, 273-278 of the manuscript. In fact, we think that the schema shown in the original version of Figure 1d might have been confusing with this respect since the neutral condition was shown next to the visual high and low value conditions, as if it also represented a certain stimulus-reward association. We have now separated the neutral condition from the rest of the conditions in Figure 1d and added an explanation about the reward associations to the figure caption that clarifies this issue (lines 307-310). 

We note, however, that including a neutral auditory condition would have produced a 3-by-2 balanced design - for three levels of reward (high, low and neutral) and two modalities (visual and auditory). We decided not to use this design based on the following reasons: Firstly, in our pilot experiments, participants had difficulties in identifying different auditory tones and in discriminating them based on their pitch. Therefore, having three rather than two different auditory tones would have made the learning of the stimulus-reward associations more difficult for the auditory stimuli. This is different from visual conditions where the discrimination of different stimuli based on their color and therefore the learning of stimulus-reward associations was relatively easy for the participants. Secondly, the main aim of the study was to characterize the effect of reward magnitude, which we could undertake by comparing the high versus the low reward condition of each modality. Towards this aim, testing the neutral condition was unnecessary. Nevertheless, the rationale for including a neutral condition was to have a setting, in which the responses to the visual target can be characterized, as we did for the results shown in Figure 3-5, without any influence of the reward associative learning. Importantly, the neutral condition consisted of a Gabor patch plus an overlaying circle as in the other conditions (see Figure 1), thus allowing to estimate the baseline visual discrimination performance and the evoked responses by the visual target in each participant. Thirdly, the perceptual discrimination threshold for all participants was determined by using stimuli that were identical to the neutral condition, i.e. a Gabor patch plus an overlaying grey circle. This again shows that our neutral condition basically represented the responses to the visual target whereas intra-modal (visual) and cross-modal (auditory) conditions represented how this response was changed by another dimension: i.e., the reward associated with each color or sound pitch. 

In general, we agree with the reviewer that visual conditions were more probable than the auditory conditions and hence the responses to them might have differed from the less frequent auditory conditions. We note, however, that each visual or auditory reward condition was compared against its counterpart with a different reward magnitude. For instance, while the responses to all the intra-modal (visual) stimuli might have been stronger or weaker than the cross-modal (auditory) stimuli because of the expectation effect that the reviewer mentioned, the difference between the high and low reward conditions of each modality is not likely to be affected by this factor. 

- The authors analyzed data using RM ANOVAs with Reward value (high, low) as one of the two factors. However, the study design included a neutral condition which has not been compared to the reward conditions. I was wondering why the authors did not include a Condition factor in their analyses with three levels (high reward, low reward, neutral).

Thanks for this comment. As mentioned in response to the first point, the neutral condition was never associated with any reward value. Therefore, its inclusion in our analyses would deter from our main aim which was to characterize the interaction between the reward value and the sensory modality. 

Due to the considrations mentioned in response to point 1, our design did not include a neutral condition for the cross-modal stimuli. Hence, the RM ANOVA that the reviewer suggested could only be applied to the data of visual (intra-modal) conditions. However, the main aim of RM ANOVAs employed in this study was to test the effect of reward value (high, low) across different modalities (intra-modal and cross-modal). The 2 by 2 RM ANOVAs that we used is therefore the most straightforward and parsimonious statistical test to examine whether the effect of reward value differed between the intra-modal (visual) and cross-modal (auditory) conditions. 

- The authors normalized data to remove the effect of perceptual biases that subjects may have for different colors or tone frequencies. This is, in my opinion, a correct procedure since there is evidence that, for instance, RT may be influenced by stimulus color even for isoluminant stimuli. However, the normalization procedure does not look appropriate to me. If I correctly understood, the authors subtracted the data of each condition in pre-conditioning from its counterpart in post-conditioning. This procedure allows to select the effect of reward but it does not eliminate the bias for specific chromaticies or tone frequencies. In my opinion, a possible correct method, given the unbalanced design, would be identifying each subjects bias in pre-conditioning (e.g., the difference between orange and blue in RTs, or the difference between high and low tone frequency) and correct the post-conditioning data by adding or subtracting the difference observed in pre-conditioning to limit if not eliminate the color (or pitch) bias. For instance, if a subject in pre-conditioning is 6 ms faster in the orange compared to the blue cue condition, 6 ms could be added in the orange condition post-conditioning data. In this way, the remaining difference between orange and blue in post-conditioning may be due to different reward value (plus noise). After this correction, the authors could implement the ANOVAs as defined.

Thanks for this comment. Please note that the association of colors and tone pitches with rewards was counter-balanced across participants, as mentioined in lines 264-267 of the manuscript and the legend to Figure 1. Specifically, we eliminated the effects of the physical characteristics of reward cues by counterbalancing colors (orange-O, blue-B) and sounds (350 Hz-L, 1050 Hz-H) across participants. Accordingly, there were 4 groups: (1) high-value O/L, low-value B/H, (2) high-value B/H, low-value O/L, 3) high-value B/L, low-value O/H, 4) high-value O/H, low-value B/L. 

In addition to the counterbalancing across participants, we additionally removed the potential effect of different colors and sound pitches by subtracting the data of pre-conditioing from post-conditioning. The procedure that we use is in fact statisticallay and numerically identical to the reviewer’s suggestion as explained in the example below: suppose that for a certain participant, RTs in response to the orange and blue colors were 300 ms and 400 ms, repectively, during the pre-conditioing. Therefore, here the participant is 100 ms faster in orange condition before the reward associations are learned. Suppose also that the RTs changed to 250 ms and 380 ms in the post-conditioing: here the participant is 130 ms faster in orange condition. Subtracting the data of pre- from post-conditioing adjusts the difference of orange versus blue from -130 ms in post-conditioing to -30 ms ((250-300) – (380-400)). So the 130ms-faster reaction time in orange is rectified with the adjustment to 30ms-faster RT, which is exactly the way that the reviewer suggested.

MINOR CONCERNS

- The experimental procedures section requires more details. For instance, the authors did not explain what kind of calibration procedure they used and for how long it lasted. Similarly, they did not specify the number of sessions they used in the QUEST method as well as the number of trials. As for the conditioning phase, it is not clear what happened after a wrong response. What kind of feedback did the participants receive in this situation? Finally, did the authors checked for data normality?

We thank the reviewer for pointing these out. We have now added these details to the Methods section. Specificially: 

Re. Calibration (lines 196-199): We have added that “An experimental session started with a calibration procedure, where for each participant the luminance of two consecutively presented colors was adjusted until the perceived flicker between them was minimized and they became perceptually isoluminant (total duration of calibration 5min). The calibration was followed by a short training session for the orientation discrimination task (Number of trials = 36)”.

Re. QUEST: We have now clarified that QUEST is an adaptive method that tries to find the best estimation of each participant’s psychophysical threshold by estimating the probability of each response given the stimulus intensity. Hence, the number of trials differed between the participants and depended on the variance of their responses. We have now included a short description of the QUEST method to lines 201-205. 

Re. Conditioning: Participants’ performance in this task was near perfect as mentioned in lines 392-394. In the rare event of errors, the feedback display informed the participants that the reward on that certain trial was zero (no feedback was provided regarding the accuracy of the decisions in any part of the experiment). We have now added this information to lines 242.

Re. Assumption of normality for parametric tests. We have now explained our procedure in lines 373-376: “Before applying the RM ANOVAs, the assumption of normality was confirmed by inspecting the histograms and Quantile-Quantile (Q-Q) plots of the data. We only observed small deviations from normality in some cases and therefore decided to proceed with our preregistered analysis plan”. 

- Rows 359-360: please report p-values of t-tests.

Thanks for pointing this out. In fact, the p-values were mentioned in the subsequent lines but for better clarity, we now provide them right after the description of the effects in each individual modality (now lines 399-404).

- Rows 392-394: the authors stated that auditory high-value cues tended to slow down N1 responses. I think the analysis does not support this statement since the p-value (0.27) is more compatible with a lack of difference. We usually indeed refer to a trend for p-values comprised between 0.05 and 0.10.

Many thanks for pointing this out. Our choice of the word “tended” was a poor choice and might have implied that we consider this effect a “statistical trend”, which was not intended. As mentioned in the preceding lines (now 430-432), we observed an interaction between Reward and Modality factors in their effect on the latency of the N1 responses: visual cues significantly sped up the N1 responses (p = 0.016) and auditory cues slowed down the responses (p = 0.27), but the latter effect did not reach statistical significance. We have now rephrased this part (lines 434-437). 

- In general (e.g., Table 2), the authors used acronyms such as IH and CL also for the pre-conditioning conditions. This may be confusing for the reader since the cues were not conditioned yet. Perhaps, I would clarify this aspect in the manuscript.

Thanks for this comment. While we understand that these acronyms may be confusing, since in pre-conditioing they refer to the stimulus conditions that were not yet associated with the rewards, we decided to keep them. This is to underscore that our effects in post-conditioning were all corrected relative to their counterparts in pre-conditioing. To clarify the choice of acronyms for the readers, we have now added the following explanation to the Methods section (lines 265-273):

“The association of each cue identity with the reward value was counterbalanced across participants. This means that each reward condition (high or low), comprised an equal number of instances where either the orange or the blue color or the 350 Hz or the 1050 Hz tone was associated with that reward magnitude. Therefore, when describing the effects during the orientation discrimination task, we will refer to all stimulus conditions with respect to the reward assignment that they acquired after the conditioning, although during the pre-conditioning these associations were not learned yet. Accordingly, these conditions in both pre- and post-conditioning will be referred to as: Intra-modal High Reward: IH, Intra-modal Low Reward: IL, Cross-modal High Reward: CH and Cross-modal Low Reward: CL, plus a neutral condition (referred to as Neut), which was never associated with any reward value (as shown in Figure 1).”

- Figure 3E: what does the double asterisk mean?

The double asterisk refers to the interaction effect between reward and modality and the post-hoc followup test for this effect, which showed a significant effect of reward only in the cross-modal condition. We have now clarified the meaning of these asterisks in Figure 3E by referring to them in the main text where the statistical tests are described (Lines 468 and 471).

- FIGURES 2,3,5: the authors are using a color code for bars which is not defined in the captions.

The color codes in Figure 2 are defined (red for high and blue for low reward) in the legend. In Figure 3 and Figure 5, color codes correspond to condition labels shown on the x-axis, which are described in the legend (lines 486-487 and lines 597-599 for Figure 3 and 5, respectively). 

- The manuscript requires some editing. References are not consistently formatted (e.g., row 73) and there are many language mistakes (e.g., ‘auditory tones’, row 177: ‘different’ instead of ‘difference’) and typos (e.g., caption Figure 5: ‘pre-condoning’; row 645: ‘enchantment’; row 664: ‘at a the’)

Thank you for pointing these mistakes out. We have now corrected all formatting issues and language mistakes. 

Comments of Reviewer 2

Reviewer #2: In this very interesting study, the authors aimed to investigate the effect of modality (acustic or visual) and value of a reward in an orientation discrimination task, using both psychophysical and electrophysiological measures. They found that a cross-modal highly rewarded cue is efficient in improving visual sensitivity, while intra-modal reward cues tend to interfere with a visual task.

Thank you for the positive evaluation of our work and for the extremely helpful suggestions, which we have carefully implemented as outlined below.

There are some concerns about the experimental design and the statistical analysis.

MAJOR CONCERNS:

- The authors comment continuously not significant data, and I think this can be correct when this is underlined. When you read the article you have the impression of solid results, but at the end of many paragraph you discover that we are talking about trends. Even, on line 394, we read of a trend only to discover that the p value is 0.27. This is clearly not a trend. This is particular evident also in lines 420-437. I ask the authors, to make everything clearer, to break down the significant effects identified and talk about them first, and then proceed to identify the trends, and then reformulate everything.

Many thanks for this helpful comment. We agree that we should have separated the signficiant and insignificant effects more clearly. We have now amended this problem by clearly stating which effects were statistically significant, correcting reporting errors due to a poor choice of words (p = 0.27, also see the response to reviewer 1 about this), and clarification of the statistical trends which did not reach significance. Accordingly, we have applied changes throughout the manuscript (including corrections to the abstract and the discussion, see for instance lines 428-429 and 462 of the Results, lines 642-643, 668-669 and 670-673 of the Discussion).

- The value of reward is not well balanced in the visual and auditory modality: there are three values for the visual cues (high-neutral and low value) and only two for the auditory cues (low and high). This is an important issue to me, because the salience of visual cues may be decreased for this very reason. The problem is that the pejorative effect of visual cues could be due to this aspect of the experimental design. Authors should therefore at least explain why they used a neutral condition, why they also didn't present the data in the neutral condition, or better, describe also this condition (neutral visual cue). Naturally, they should convince readers that the pejorative effect of visual cues is not due to this factor.

Thanks for this comment. We think that the schema presented in Figure 1d might have been misleading, giving the impression that the neutral condition is equivalent to a reward value equal to 0. In fact, the neutral condition was never presented during the conditioning phase and the participants did not associate the neutral cue with any reward. This important point is now clarified in lines: 226-227, 231-233 and 273-278. We have also corrected Figure 1d and the legend to this panel to clarify this issue. As explained in lines 273-278 of the revised manuscript (and the lengend to Fig.1E), the rationale to include the neutral condition was to have a setting in which the responses to the visual target can be characterized. Since the neutral condition consisted of a Gabor patch plus an overlaying circle, similar to all other conditions (see Figure 1), inspection of the results in neutral condition allowed us to have an estimation of the baseline visual discrimination performance and the evoked responses by the visual target in each participant, as shown in Figure 3-5, (which was otherwise impossible as the visual target and the reward cues were at the same spatial location). 

As also mentioned in response to Reviewer 1’s first comment, we agree with the reviewer that visual conditions were more probable than the auditory conditions and hence the responses to them might have differed from the less frequent auditory conditions. We note, however, that each visual or auditory reward condition was compared against its counterpart with a different reward magnitude. For instance, while the responses to all intra-modal (visual) stimuli might have been stronger or weaker than cross-modal (auditory) stimuli because of the expectation effect that the reviewer mentioned, the difference between high and low reward conditions of each modality is not likely to be affected by this factor. 

Due to these reasons we do not think that our reported intra-modal value effects are related to the neutral condition.

MINOR CONCERNS:

- There are many confusing aspects: acronyms, such like IH, CH, etc could be confusing. The experimental design should be described with more clarity. Figure 1c doesn't help (I suggest to modify or eliminate it). The pitch function is not immediately clear, because at rows 211-213 the reader has to infer it. line 225: maybe it is "cues", not "stimuli". There are also some typos to be addressed.

Thank you for these very helpful suggestions that are now implemented in the revised manuscript. We have tried to increase the clarity of the manuscript and improve our description of the experimental design at multiple locations including but not limited to those suggested by the reviwer. We have also corrected some unintended errors (the topoplots in Figure 4A were swapped between the pre- and post-conditioing, which is now corrected). In the light of Reviewer 1’s comments, we decided to keep Figure 1d and add clearer explanations about our procedures to the legend. We have also corrected all the typos and linguistic mistakes, as far as we could see.

---

## [Decision Letter · Decision Letter 1]

24 May 2023

PONE-D-22-27111R1Differential effects of intra-modal and cross-modal reward value on perception: ERP evidencePLOS ONE

Dear Dr. Pooresmaeili,

Thank you for submitting your manuscript to PLOS ONE. After careful consideration, we feel that it has merit but does not fully meet PLOS ONE’s publication criteria as it currently stands. Therefore, we invite you to submit a revised version of the manuscript that addresses the points raised during the review process.

We look forward to receiving your revised manuscript.

Kind regards,

Nicola Megna, M.D.

Academic Editor

PLOS ONE

Journal Requirements:

Reviewers' comments:

Reviewer's Responses to Questions

**Comments to the Author**

1. If the authors have adequately addressed your comments raised in a previous round of review and you feel that this manuscript is now acceptable for publication, you may indicate that here to bypass the “Comments to the Author” section, enter your conflict of interest statement in the “Confidential to Editor” section, and submit your "Accept" recommendation.

Reviewer #2: All comments have been addressed

Reviewer #3: (No Response)

2. Is the manuscript technically sound, and do the data support the conclusions?

Reviewer #2: Yes

Reviewer #3: Yes

3. Has the statistical analysis been performed appropriately and rigorously? 

Reviewer #2: Yes

Reviewer #3: Yes

4. Have the authors made all data underlying the findings in their manuscript fully available?

Reviewer #2: Yes

Reviewer #3: Yes

5. Is the manuscript presented in an intelligible fashion and written in standard English?

Reviewer #2: Yes

Reviewer #3: Yes

6. Review Comments to the Author

Reviewer #2: The authors have fully answered the doubts raised in my previous review. It is very clear now what they meant by "neutral condition" and that the related cue was not associated with any reward. The meanings of the various acronyms have also been clarified in an appropriate manner and also the statistical significance of the various results is now much more readable and understandable.

Reviewer #3: This is a very interesting study investigating the effect of intra-modal and cross-modal value on visual perception and its neural correlates combining psychophysics and EEG. I have been involved in the review process after a first round of review, after the authors already addressed several comments. The study investigates an interesting question, using a solid experimental paradigm. The results are interesting and somehow unexpected, with potentially important implications for our understanding of how the sensory brain encodes reward value. The analysis of the data is sound, and the manuscript is overall well written.

I only have some minor comments:

1-Since the main EEG results are based on the difference between pre and post-conditioning traces (which I think is a very intellingent way of analysing the data), I think it might be helpful to show two separate panels (perhaps in figure 5), the average trace of the difference between post and pre conditioning for HI LI and HC and LC conditions. I think that this would help the reader to appreciate better the results, to complete the information provided by the small maps at the bottom of Figure 4b and c.

2- I find the results that, during the conditioning phase, auditory stimuli elicit a stronger P1 component (larger amplitude and shorter latency) than visual stimuli very intriguing. Could the authors discuss this result? How do they interpret it?

3- During the conditioning phase, is there a difference between the first and the second half of the trials? If participants learn the association between a particular stimulus and either high or low reward, a difference between the two migh emerge later on during the experimental session.

4-Figure 3e-f and Figure 5c: individual subjects data could be overlayed on top of the bar plots so to better appreciate the interindividual variability of the effect.

line 124: Figure 4d actually refers to Figure 2b, please correct the typo.

7. PLOS authors have the option to publish the peer review history of their article (what does this mean?). If published, this will include your full peer review and any attached files.

Reviewer #2: **Yes: **Nicola Megna

Reviewer #3: No

---

## [Author Response · Author response to Decision Letter 1]

10 Jun 2023

Point-by-point response

Comments of Reviewer 2

Reviewer #2: The authors have fully answered the doubts raised in my previous review. It is very clear now what they meant by "neutral condition" and that the related cue was not associated with any reward. The meanings of the various acronyms have also been clarified in an appropriate manner and also the statistical significance of the various results is now much more readable and understandable.

We thank the reviewer for the very insightful suggestions during the first round of revisions which have greatly helped us to improve our manuscript and make it much stronger than the initial submission.

Reviewer #3: This is a very interesting study investigating the effect of intra-modal and cross-modal value on visual perception and its neural correlates combining psychophysics and EEG. I have been involved in the review process after a first round of review, after the authors already addressed several comments. The study investigates an interesting question, using a solid experimental paradigm. The results are interesting and somehow unexpected, with potentially important implications for our understanding of how the sensory brain encodes reward value. The analysis of the data is sound, and the manuscript is overall well written.

Thank you for the positive evaluation of our work and for the extremely helpful suggestions, which we have carefully implemented as outlined below.

I only have some minor comments:

1-Since the main EEG results are based on the difference between pre and post-conditioning traces (which I think is a very intellingent way of analysing the data), I think it might be helpful to show two separate panels (perhaps in figure 5), the average trace of the difference between post and pre conditioning for HI LI and HC and LC conditions. I think that this would help the reader to appreciate better the results, to complete the information provided by the small maps at the bottom of Figure 4b and c.

Thank you for this great suggestion. Since both Figure 4 and 5 already contain many dense panels, we decided to add the difference waves (for HI-LI and HC-LC) and their corresponding topographic distributions to the Supplementary Information (Supplementary Figure 4) and refer to it in the legend of Figure 4 and in the main text (lines 545 and 562). 

2- I find the results that, during the conditioning phase, auditory stimuli elicit a stronger P1 component (larger amplitude and shorter latency) than visual stimuli very intriguing. Could the authors discuss this result? How do they interpret it?

Many thanks for pointing this out. Auditory tones have overall shorter processing latencies compared to visual stimuli [1]. However, as the reviwer noted, it is indeed intriguing that in visual cortex, an area that is dedicated to the processing of visual stimuli, the amplitude of P1 responses is larger for auditory compared to visual stimuli. We think that this finding is due to the relative strength of the visual and auditory stimuli that we utilized: visual stimuli in our experiments were small, transparent colored circles (0.44° in diameter), whereas the auditory stimuli were played at a suprathreshold intensity (70 dB) so that participants could detect the tones very easily. In fact, a different approach would be to equalize the perceived intensity of the auditory and visual stimuli by measuring participants’ detection threshold in a 2AFC paradigm for both stimulus types, similar to previous psychophsical studies [2]. However, in our experience (unplublished data), participants can easily detect auditory stimuli even at very low intensities and therefore using an equalization method for visual and auditory stimuli may yield extremely low intensities for visual stimuli which would make the discrimination between them difficult. Since we measure response latencies and amplitudes of P1 and N1 for each modality separately for all phases of the experiment and correct the data of each condition relative to its pre-conditioing counterpart during the test phase, we do not believe that the pattern of results we report in the test phase is affected by the difference in visual and auditory reward cues intensity. This is however a possibility that could be explored in future studies. We have now added these explanations to line 738-746 of the revised manuscript:

“In this vein, our results also indicated that the perceived intensity of visual and auditory stimuli might have been different as both P1 and N1 components were stronger for auditory compared to visual stimuli during the conditioning phase. Since we quantified the ERP measures separately for each sensory modality, calculated the reward effects against stimuli from the same modality, and corrected for pre-conditioning biases, it is unlikely that our reported results are due to the differences in perceived intensity of auditory and visual stimuli. We note however that using an adaptive method to equalize the perceived intensity of auditory and visual stimuli would be an interesting addition to future studies.”

3- During the conditioning phase, is there a difference between the first and the second half of the trials? If participants learn the association between a particular stimulus and either high or low reward, a difference between the two migh emerge later on during the experimental session. 

This is a great point. In order to test whether our results during the conditioning (reaction times, P3 amplitude and N1 latencies) are affected by the learning of reward associations which evolves in time, we repeated each analysis by including the phase of learning (i.e., first or second half of conditioning) as a factor. We did not observe a significant interaction between any of the reported effects and the phase of conditioning (first versus second half). These results (below) are now presented in lines 448-457 of the revised manuscript:

“Since learning of reward associations may take time and behavioral and ERP effects of reward could only arise after associative learning has been completed, we wondered whether our reported results differed between the first and the second half of the conditioning. To test this possibility, we divided the trials for each condition to two halves and entered an extra factor, i.e., phase (first or second half of conditioning) in all our ANOVAs. We found similar effect sizes in each case (the interaction between reward and modality for RT: F(1,35) = 7.70, p = 0.009, ηp2= 0.18 and for N1 latency: F(1,35) = 7.26, p = 0.011, ηp2= 0.172, and the main effect of reward on P3 amplitude: F(1,35) = 4.27, p = 0.046, ηp2 = 0.109), but we did not observe a significant interaction with the phase (all ps0.1). Therefore, our reported results did not show a dependence on the phase of conditioning, likely because full learning of reward associations was achieved very fast.”

4-Figure 3e-f and Figure 5c: individual subjects data could be overlayed on top of the bar plots so to better appreciate the interindividual variability of the effect.

Thanks for this suggestion. We have now added the individual data to both figures (3e-f and 5c).

line 124: Figure 4d actually refers to Figure 2b, please correct the typo.

Thank you for brining this error to our attention (it was in line 424). We have now corrected the typo (from Figure 4d to Figure 2b). 

1. King AJ. Multisensory Integration: Strategies for Synchronization. Curr Biol. 2005;15: R339–R341. doi:https://doi.org/10.1016/j.cub.2005.04.022

2. Ragot R, Cave C, Fano M. Reciprocal effects of visual and auditory stimuli in a spatial compatibility situation. Bull Psychon Soc. 1988;26: 350–352. doi:10.3758/BF03337679

---

## [Editor Report · Decision Letter 2]

15 Jun 2023

Differential effects of intra-modal and cross-modal reward value on perception: ERP evidence

PONE-D-22-27111R2

Dear Dr. Pooresmaeili,

We’re pleased to inform you that your manuscript has been judged scientifically suitable for publication and will be formally accepted for publication once it meets all outstanding technical requirements.

Kind regards,

Nicola Megna, M.D.

Academic Editor

PLOS ONE